# SWE-RM: Execution-Free Feedback for Software Engineering Agents

**Kashun Shum**[1,2*], **Binyuan Hui**[2*], **Jiawei Chen**[2], **Lei Zhang**[2], **X. W.**[2], **Jiaxi Yang**[2]
**Yuzhen Huang**[1], **Junyang Lin**[2], **Junxian He**[1]

[1] The Hong Kong University of Science and Technology, [2] Qwen Team, Alibaba Group
*Equal Contribution
{ksshumab,junxianh}@cse.ust.hk, binyuan.hby@alibaba-inc.com

## Abstract

Execution-based feedback like unit testing is widely used in the development of coding agents through test-time scaling (TTS) and reinforcement learning (RL). This paradigm requires scalable and reliable collection of unit test cases to provide accurate feedback, and the resulting feedback is often sparse and cannot effectively distinguish between trajectories that are both successful or both unsuccessful. In contrast, execution-free feedback from reward models can provide more fine-grained signals without depending on unit test cases. Despite this potential, execution-free feedback for realistic software engineering (SWE) agents remains underexplored. Aiming to develop versatile reward models that are effective across TTS and RL, however, we observe that two verifiers with nearly identical TTS performance can nevertheless yield very different results in RL. Intuitively, TTS primarily reflects the model's ability to select the best trajectory, but this ability does not necessarily generalize to RL. To address this limitation, we identify two additional aspects that are crucial for RL training: classification accuracy and calibration. We then conduct comprehensive controlled experiments to investigate how to train a robust reward model that performs well across these metrics. In particular, we analyze the impact of various factors such as training data scale, policy mixtures, and data source composition. Guided by these investigations, we introduce SWE-RM, an accurate and robust reward model adopting a mixture-of-experts architecture with 30B total parameters and 3B activated during inference. SWE-RM substantially improves SWE agents on both TTS and RL performance. For example, it increases the accuracy of `Qwen3-Coder-Flash` from 51.6% to 62.0%, and `Qwen3-Coder-Max` from 67.0% to 74.6% on SWE-Bench Verified using TTS, achieving new state-of-the-art performance among open-source models. On RL training, SWE-RM lifts the resolve rate of execution-based counterparts by 3 absolute points on SWE-Bench Verified.

## 1 Introduction

The automation of complex software development tasks through coding agents represents a significant frontier in large language models (LLMs). A critical component in developing these agents is the feedback mechanism used during training and evaluation, particularly through reinforcement learning (RL) (Wei et al., 2025; Qwen Team, 2025; Zhang et al., 2026) and test-time scaling (TTS) (Xia et al., 2024; Jain et al., 2025). Broadly, these mechanisms fall into two categories: execution-based verifiers (Xia et al., 2024; Jain et al., 2025), which rely on concrete outcomes like unit test results, and execution-free verifiers[1] (Pan et al., 2025; Luo et al., 2025), which are typically model-based reward models that provide a continuous score without sandbox environments.

While widely used, execution-based feedback has inherent limitations. It provides only a sparse, binary signal (pass/fail), which makes it difficult to distinguish between different successful or unsuccessful trajectories. Beyond this lack of granularity, unit tests require comprehensive coverage to yield accurate assessments, which is often unavailable. To address this challenge, exiting works rely

---

[1]Throughout this paper, we will use "reward model" and "execution-free verifier" interchangeably.

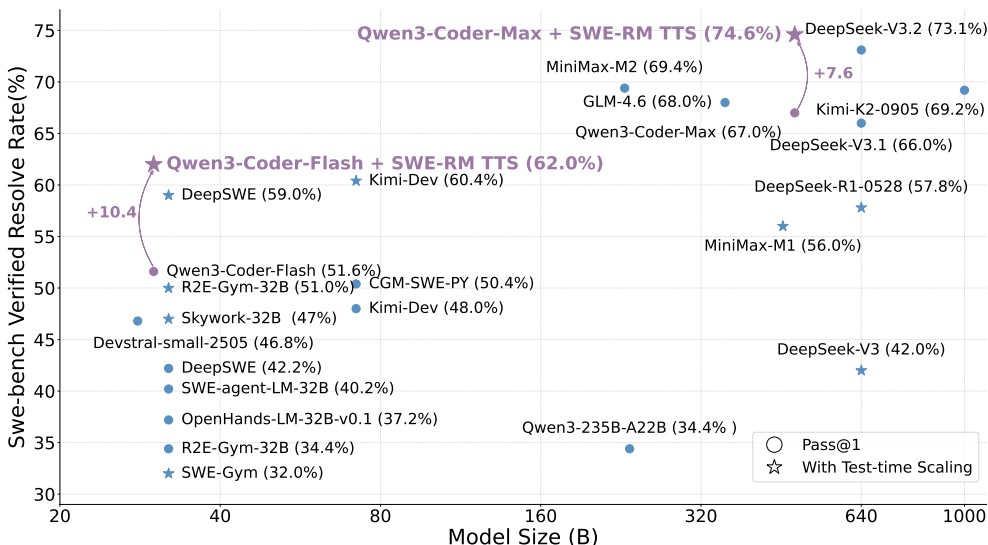

Figure 1: The pass@1 and TTS resolve rate of various open-source and proprietary models on SWE-Bench Verified. Part of the baseline results are referenced from He et al. (2025).

on extracting unit test from Github repos (Jimenez et al., 2024) or model-generated unit tests (Yang et al., 2025; Jain et al., 2025) that are not rigorously validated. For example, in issue-fixing tasks, the unit tests used from real GitHub repositories are often overly specific, and in some cases, entirely unrelated to the target issue (OpenAI, 2025). As a result, execution-based feedback limits the code data that can be used for effective reinforcement learning or test-time scaling due to requirements of high-quality unit tests. When such tests are unreliable, the resulting feedback becomes a significant challenge for RL, where nuanced and consistent reward signals are essential. Execution-free feedback offers a compelling alternative by providing continuous, fine-grained scores across entire trajectories, allowing for better discrimination among candidate solutions and reducing bias toward specific patches. Despite its promise, execution-free feedback remains largely underexplored, and its properties in the context of SWE agents are not yet well understood.

In this work, we aim to develop a versatile and effective reward model usable across different scenarios such as TTS and RL for software engineering. While it is straightforward to adopt TTS (e.g., best of $k$) performance directly as the metric to guide the reward model training (Pan et al., 2025), our initial findings reveal that two verifiers with nearly identical TTS performance can show drastically different behavior in RL. This leads us to a fundamental research question: *What properties determine a reward model's effectiveness in RL training, and how can we develop an all-round SWE reward model that performs well in both TTS and RL?*

Intuitively, TTS primarily measures a verifier's ability to rank the correct solution highest among multiple candidates, but it overlooks aspects that are essential for RL: the ability to effectively distinguish correct from incorrect trajectories and to produce scores that reliably correspond to the degree of correctness. Based on this observation, we further evaluate reward models using additional metrics: AUC, which reflects the correctness of relative ordering across trajectories, and ECE (Guo et al., 2017), which measures calibration representing whether the verifier's scores align with empirical correctness. We demonstrate that AUC and calibration provide complementary information to TTS and are both critical for ensuring the reward model delivers reliable signals in RL.

To train a reward model that performs well across these metrics, we conduct large-scale ablation studies examining the effects of training data scale, the ratio of positive to negative samples, mixtures of data sources, and context length. These investigations lead to a practical recipe for building robust, execution-free reward models tailored to SWE tasks. Guided by these investigations, we obtain **SWE-RM**, an accurate and robust reward model with 30B total and 3B activated parameters for advancing SWE agents. On SWE-bench Verified (OpenAI, 2025), SWE-RM lifts the accuracy of `Qwen3-Coder-Flash` from 51.6% to 62.0% and `Qwen3-Coder-Max` from 67.0% to 74.6%, achieving best-in-class among 30B-level and all open-source models respectively, as shown in Figure 1. Moreover, SWE-RM is highly effective when used as a reward signal in agentic RL training.

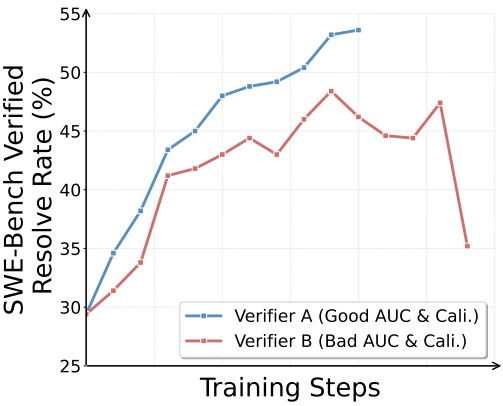 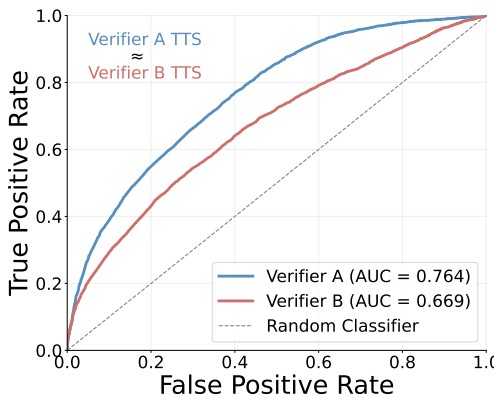

Figure 2: RL training curves of two verifiers with similar TTS performance. Despite comparable TTS, the downstream RL outcomes differ drastically.

Figure 3: Two models with similar TTS performance (Model A with +4.7% and Model B with +4.5%) show significant differences in their AUC scores.

For example, it improves the RL performance of execution-based counterparts by 3 absolute points on SWE-bench Verified.

## 2 RELATED WORK

In Software Engineering (SWE) tasks, verifiers fall into two categories: execution-based, which rely on unit tests (e.g., Agentless (Xia et al., 2024), R2E-Gym (Jain et al., 2025), DeepSWE (Luo et al., 2025)), and execution-free, which use model-based scoring (e.g., SWE-Gym (Pan et al., 2025), OpenHands Critic (Team, 2025)). Existing work on execution-free verifiers has primarily emphasized test-time scaling (Pan et al., 2025; Jain et al., 2025; Luo et al., 2025), with limited attention to other quality dimensions. We show that an execution-free verifier's quality also depends on classification accuracy and calibration, and provide a systematic study of training such verifiers. In reinforcement learning for SWE agents (e.g., OpenHands (Wang et al., 2025), SWE-Agent (Yang et al., 2024)), execution-based feedback—akin to rule-based metrics in math (DeepSeek-AI, 2025)—has enabled recent model training (e.g., Qwen3-Coder (Qwen Team, 2025), GLM-4.5 (Team et al., 2025a), MiniMax-M1 (MiniMax, 2025)), but is constrained by noisy test suites and sparse signals. We are the first to integrate execution-free feedback into SWE agentic RL, demonstrating its potential to deliver finer-grained rewards and improve efficiency. An extended related work are discussed in Appendix B.

## 3 WHAT DEFINES A VERSATILE REWARD MODEL FOR SWE?

Our goal is to develop a versatile reward model that performs well across both TTS and RL. Following common practice, we begin by examining whether TTS performance can serve as a reliable guide for selecting a reward model for RL. We first present our initial findings that highlight the limitations of relying solely on TTS performance to guide reward model training. This result demonstrates that TTS alone cannot explain downstream success in RL, raising important questions about what properties of a verifier matter. To resolve this gap, we next revisit the role of TTS as an evaluation metric, analyze its limitations, and introduce complementary criteria—AUC and calibration—that provide a more holistic view of verifier quality.

### 3.1 INITIAL FINDINGS: LIMITATIONS OF RELYING SOLELY ON TTS

As we aim to develop a versatile reward model that can be applied across different scenarios such as TTS and RL, but it is unknown what defines such a versatile reward model and whether TTS and RL impose different requirements. A natural question to ask is whether a reward model that performs well on TTS will also perform well on RL. To avoid ambiguity, the reward model is trained purely through supervised next-token prediction instead of using TTS as the training signal. TTS

is used exclusively as an evaluation metric for model selection etc. More training details will be introduced in § 4.1. Our initial exploration reveals an intriguing finding: two execution-free verifiers that achieve nearly identical TTS improvements give rise to strikingly different behaviors when used as reward models in reinforcement learning. As shown in Figure 2, both verifier A and verifier B achieve similar TTS improvements, indicating, at first glance, that they are equally effective at choosing the correct solution highest among candidate trajectories. Yet, when deployed in RL training, verifier A supports smooth improvement, while verifier B exhibits significant instability, failing to provide reliable learning signals and eventually causing RL training to collapse.

This result challenges the widely adopted view of TTS as a sufficient proxy for verifier quality (Lightman et al., 2023; Pan et al., 2025). If two models are judged equivalent by TTS but behave so differently in RL, then TTS alone cannot capture the aspects of a verifier that truly matter for reinforcement learning. In other words, TTS provides only a partial picture: it summarizes top-1 ranking ability but hides other properties that directly affect how reward signals shape policy updates. These findings prompt us to ask a fundamental research question: *What properties determine a reward model's effectiveness in RL training, and how can we develop an all-round SWE reward model that performs well in both TTS and RL?*

To answer this, we must carefully reconsider what TTS actually measures, why it fails to explain the RL discrepancy observed, and what alternative metrics can reveal the missing dimensions of verifier quality. This motivates our shift beyond TTS to a broader and versatile evaluation.

## 3.2 MOVING BEYOND TTS: THE NEED FOR MORE VERSATILE EVALUATION ON VERIFIER QUALITY

At first glance, TTS appears to be a natural metric: it checks whether the correct solution trajectory is ranked highest among a pool of candidates. Intuitively, this reflects a verifier's ability to make the right top-1 decision, however, TTS only measures a narrow slice of verifier capability. By focusing exclusively on whether the single best trajectory is ranked first, TTS ignores two properties that become critical once the verifier is used as a reward model in reinforcement learning. The first overlooked dimension is *discriminative ability*. In RL, the agent generates a wide range of trajectories, some of which are unresolved. The verifier must provide accurate feedback not just for the best trajectory, but across many near-miss or partially correct candidates. A verifier with weak discriminative ability will assign similar scores to both correct and incorrect trajectories, producing noisy reward signals that compromise policy updates. The second is *confidence reliability*, or calibration. In RL, verification scores are often interpreted as proxies for the likelihood of correctness, serving as the reward magnitudes used to guide policy learning. If these scores are mis-calibrated, for instance, a normalized score of 0.9 reflects only a 60% probability of being actual correct—then the policy receives misleading signals about the expected value of its actions. Poor calibration can therefore poison the reward shaping process, leading to unstable or collapsed training dynamics even if top-1 accuracy (TTS) appears satisfactory.

These overlooked dimensions provide a natural explanation for the discrepancies we observed in Figure 2, and to capture these dimensions, we supplement TTS with two complementary metrics. AUC (Bradley, 1997) evaluates discriminative ability by measuring how well the verifier separates resolved from unresolved trajectories across the entire distribution, rather than focusing only on the best. Calibration (Wang, 2025) quantifies the alignment between predicted confidence scores and empirical correctness, for example by using Expected Calibration Error (ECE) (Guo et al., 2017). A higher AUC means models can better descriminate resolved and unresolved trajectories while a lower ECE means there is lower mismatch between confidence and accuracy, indicating higher reliability. Together, these three metrics—TTS, AUC, and calibration—form a more versatile evaluation toolkit: they jointly capture top-1 ranking accuracy, overall discriminative power, and reliability of confidence estimates.

Empirical analysis demonstrates the importance of considering all three. **(1) Discriminative gap despite equal TTS:** As shown in Figure 3, verifier A and verifier B obtain nearly identical TTS improvements (+4.7% vs. +4.5%), yet their AUC scores differ by 0.095. Thus, although both appear equivalent by TTS, only verifier A reliably distinguishes resolved from unresolved trajectories—a property crucial for producing consistent reward signals. **(2) Calibration and score distribution disparity:** Figure 4 reveals that verifier B suffers from widespread over- and under-confidence,

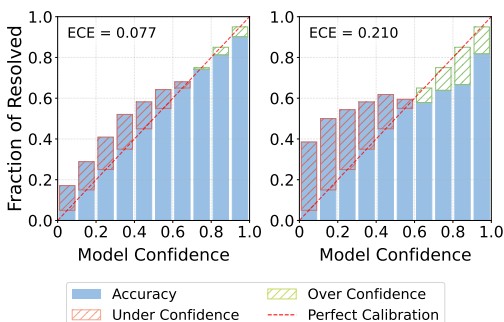 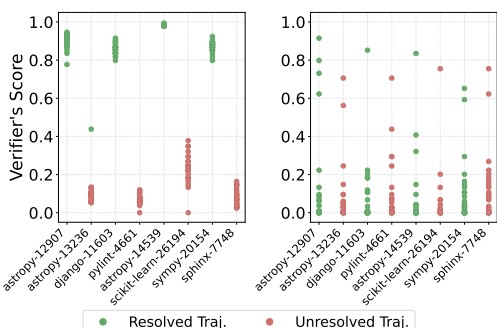

Figure 4: Reliability diagrams for verifier A (left) and verifier B (right) with similar TTS performance.

Figure 5: Score distribution cases for verifier A (left) and verifier B (right) with similar TTS performance.

while verifier A is three times better calibrated according to expected calibration error. Again, TTS fails to reflect this difference, though it directly affects the trustworthiness of reward magnitudes. Figure 5 further highlights why this matters: across 32 runs of random selected 8 instances, Model A consistently assigns high scores to resolved trajectories and low scores to unresolved ones, making it possible to set a reliable threshold for acceptance. In contrast, Model B frequently assign unexpectedly low scores for resolved trajectories, while unresolved trajectories receive inflated scores—producing overlapping distributions that might confuse policy models. The trajectories used for calibration and score distribution analysis are sampled from `Qwen3-Coder-Max` while the other settings are same as the one we will show in Appendix D.1. This combination of miscalibration and poor separation explains why Model A provides misleading signals in RL even when its TTS remains competitive. We also include a theoretical analysis between the three metrics and RL dynamics in Appendix C.

These observations underscore that TTS, accuracy, and calibration are complementary metrics, each capturing a distinct aspect of verifier capability. The discussion of metrics in this section is limited to using early, small-scale RM variants as illustrative examples. And in subsequent investigation—beginning after we motivate the need for AUC and calibration—marks the start of the actual supervised RM training and large-scale, comprehensive ablations. We will show how to obtain a versatile and robust reward model in § 4 and discuss the implications for reinforcement learning in § 5.

## 4 HOW TO TRAIN A VERSATILE REWARD MODEL FOR SWE?

To build a versatile and robust reward model as discussed in § 3, we conclude and analyze several critical factors that significantly influence final performance. Specifically, we systematically investigate training data scale, the ratio of positive to negative samples, policy, data source, and context length, and discuss their impact on the verifier's three core abilities. These observations collectively guide the development of SWE-RM, which achieves superior performance.

### 4.1 TRAINING METHODS

Following SWE-Gym (Pan et al., 2025), we formulate reward modeling as a generative classification task, where the reward model takes a trajectory as input and outputs a special token (e.g., YES/NO). Given the full multi-turn trajectory, the model is prompted to output a single special token, either YES (resolved) or NO (unresolved). And the supervised fine-tuning utilizes standard next-token prediction loss on this special token. At inference time, by obtaining the log probability of the special token YES($l_y$) and NO($l_n$), the final score $r$ is calculated by $\exp(l_y)/(\exp(l_y) + \exp(l_n))$, which maps to a continuous reward model score $r \in [0, 1]$. To construct training data, we collect agent trajectories by deploying different policy models (`Qwen3-Coder` and `Claude-4`) to interact with the agent scaffold OpenHands (Wang et al., 2025) across multiple training data sources, including SWE-Gym (Pan et al., 2025), SWE-rebench (Badertdinov et al., 2025), SWE-smith (Yang et al., 2025), and R2E-Gym (Jain et al., 2025). These trajectories are then labeled as positive or negative based on their execution results with the provided fail2pass test.

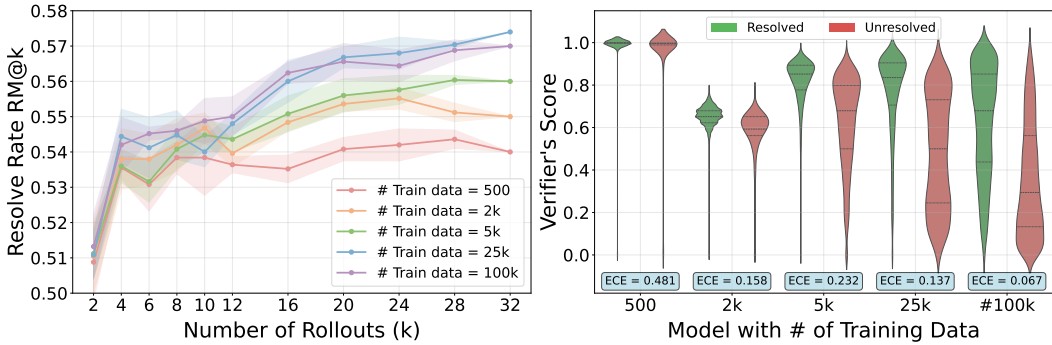

Figure 6: Left: Test-time Scaling curve of models trained with different # of training data. Right: Distribution of verifier scores across all evaluated trajectories. Clearer separation between resolved and unresolved trajectories, along with a lower ECE, indicates better performance.

**RM Training Setup** We use `Qwen3-30B-A3B` (Qwen Team, 2025) as the base model for reward model training, as it provides a balance between efficiency and strong coding capabilities. Evaluation is primarily conducted on SWE-bench Verified (Jimenez et al., 2024), a curated subset of 500 human-verified tasks designed to reliably assess model performance on real-world software engineering problems. For the evaluation metrics, test-time scaling is measured using 32 independent runs for each instance on SWE-bench Verified. Accuracy is calculated using the AUC score across all $32 \times 500$ trajectories, while calibration is assessed by the expected calibration error (Guo et al., 2017). RM@K is defined as the resolve rate of the final selected trajectories from $k$ samples. For each $k < 32$, we report the mean and variance over 5 random runs to ensure fair evaluation. Further details on the reward model training setup can be found in Appendix D.

## 4.2 DATA SCALING AND RATIO EFFECT

Poorly trained reward models often exhibit unexpected behavior when evaluated on out-of-domain (OOD) data. This OOD generalization challenge is particularly severe in SWE tasks, where multi-turn interactions create a substantially larger output space compared to traditional reasoning tasks, which might requires more training data. As shown in Figure 6, we uniformly sampled varying amounts of training data from different policy models and data sources. The left subfigure demonstrates that models trained on more than 20k samples generally achieve improved test-time scaling performance as $k$ increases, whereas models trained on fewer samples (e.g., fewer than 5k) may even experience declining performance. We attribute this to the limited generalization capacity of under-trained models: as $k$ grows, the probability of encountering OOD trajectories increases, and test-time scaling becomes highly sensitive to such cases. Even a single erroneously high score assigned to an incorrect trajectory can significantly distort the final resolve rate.

While TTS performance improves as the training data size increases up to 25k, further expansion to 100k yields diminishing returns. the score distributions in the right subfigure of Figure 6 reveal that larger training datasets enhance discriminative ability, as evidenced by clearer separation between resolved and unresolved trajectories. Moreover, models trained on more data demonstrate improved calibration: for instance, a model trained with only 500 examples has an ECE of 0.481—seven times higher than that of a model trained with 100k examples. This indicates that scaling up training data produces more reliable scores, highlighting the effectiveness of data scaling.

To further investigate the role of data composition, we fix the total amount of training data and vary the ratio of positive to negative trajectories, as shown in Table 1. We observe that across both model scales, the 2:1 ratio generally achieves the best overall performance in terms of AUC, calibration, and test-time scaling. Due to the limited availability of positive data, we experiment with ratios up to 2:1. To further investigate the role of data composition, we fix the total amount of training data and vary the ratio of positive to negative trajectories, as shown in Table 1. We observe that across both model scales, the 2:1 ratio generally achieves the best overall performance in terms of AUC, calibration, and test-time scaling. By contrast, more balanced ratios such as 1:1 avoid extreme skew but still fall short of the 2:1 configuration. Importantly, the 2:1 ratio also offers higher efficiency, as it requires a smaller pool of negative data while still utilizing all available positive data in practice.

Table 1: Effect of training verifiers with different positive-to-negative data ratios on AUC, ECE, and test-time scaling performance (best results in bold).

| Ratio | Qwen3-Coder-Flash | | | Qwen3-Coder-Max | | |
|-------|------|------|-------|------|------|-------|
| | AUC | ECE↓ | RM@32 | AUC | ECE↓ | RM@32 |
| 2 : 1 | **0.805** | **0.080** | **62.0%** | **0.755** | **0.121** | 71.0% |
| 1 : 1 | 0.782 | 0.132 | 60.8% | 0.734 | 0.157 | 70.2% |
| 1 : 2 | 0.789 | 0.235 | 61.0% | 0.736 | 0.371 | 69.4% |
| 1 : 4 | 0.789 | 0.185 | 61.6% | 0.742 | 0.299 | **71.8%** |
| 1 : 8 | 0.778 | 0.349 | 60.2% | 0.738 | 0.541 | 70.6% |

Considering this balance between effectiveness and efficiency, we adopt the 2:1 ratio as the default configuration in subsequent experiments.

## 4.3 Context Length Constraint

While previous execution-free verifiers in SWE mainly support a context length of 32k (Pan et al., 2025; Jain et al., 2025), our execution-free verifiers are the first to scale up to 256k context length, enabling the scoring of complex and long trajectories. This is especially important for challenging questions, which typically involve extremely long contexts. As shown in Table 2, only when the context length is extended to 128k can more than 99% of trajectories be successfully scored without exceeding the limit. Furthermore, as models are able to score more trajectories, execution-free verifiers achieve better test-time scaling performance, as reflected in the increasing RM@32. A more detailed discussion on context length are shown in Appendix D.4.

Table 2: Effect of Context Length Scaling on verifier's score rate and test-time scaling performance. Score rate represent how many percent of trajectories can be successfully scored without exceeding the context window.

| Context Len. | Score Rate | RM@32 |
|-------|------|------|
| 16 k | 0.5% | 66.8% |
| 32 k | 12.5% | 67.4% |
| 64 k | 88.3% | 70.6% |
| 128 k | 99.5% | 73.0% |
| 256 k | **100%** | **74.4%** |

## 4.4 Policy and Source Ablation

We also examine the impact of training data collected from different policy models on verifier performance. For on-policy data, we sample training examples using the corresponding Flash/Max model on SWE-rebench, while for off-policy data we sample using Claude-sonnet-4 (Anthropic). As shown in Table 3 policy ablation, while on-policy data sometimes yields stronger results on specific metrics (e.g., TTS on Qwen3-Coder-Max), overall the Mix-Policy setting provides a better balance across AUC, ECE, and ranking. This indicates that combining on- and off-policy data enhances the generalization ability of the verifier. Such findings also reflect the advantage of our comprehensive evaluation in revealing robust trends that TTS-only analyses might overlook.

We further investigate the impact of training data sources on verifier performance. As shown in Table 3, under single-source settings, SWE-rebench achieves the best results in both AUC and RM@32, indicating that rebench may provide the highest-quality data. However, incorporating SWE-smith and SWE-Gym leads to improved calibration (lower ECE), and adding more sources enhances data scaling effects as we shown in § 4.2. Based on these observations, our final setup uses a mixture primarily derived from SWE-rebench, supplemented with data from SWE-smith and SWE-Gym, achieving a balance between quality, calibration, and scalability.

## 5 SWE-RM: a Versatile Reward Model for TTS and RL

In this section, we present SWE-RM, an accurate and robust execution-free verifier that not only achieves state-of-the-art test-time scaling performance but also significantly improves downstream reinforcement learning. We first demonstrate the superior performance of SWE-RM in TTS (§ 5.1) and then RL (§ 5.2).

Table 3: Effect of training verifiers with different policy mixture and data source on three core abilities(best results in bold).

| METHODS / MODELS | Qwen3-Coder-Flash | | | Qwen3-Coder-Max | | |
|---|---|---|---|---|---|---|
| | AUC | ECE↓ | RM@32 | AUC | ECE↓ | RM@32 |
| *Policy Ablation* | | | | | | |
| On-Policy | 0.785 | 0.148 | 58.6 | 0.727 | **0.067** | **71.0** |
| Off-Policy | 0.778 | 0.113 | 58.2 | 0.728 | 0.145 | 70.6 |
| Mix-Policy | **0.804** | **0.033** | **59.6** | **0.751** | 0.082 | 70.2 |
| *Source Ablation* | | | | | | |
| SWE-rebench (Badertdinov et al., 2025) | **0.814** | 0.076 | **0.612** | **0.774** | 0.048 | **0.718** |
| SWE-smith (Yang et al., 2025) | 0.781 | **0.033** | 0.584 | 0.736 | 0.039 | 0.70 |
| SWE-Gym (Pan et al., 2025) | 0.776 | 0.087 | 0.588 | 0.742 | 0.044 | 0.714 |
| SWE-Gym + SWE-smith | 0.813 | 0.034 | 0.602 | 0.772 | 0.035 | 0.72 |
| SWE-Gym + SWE-rebench | 0.802 | 0.087 | 0.61 | 0.762 | 0.039 | 0.712 |
| SWE-rebench + SWE-smith | 0.807 | 0.138 | 0.596 | 0.765 | 0.107 | 0.714 |
| SWE-rebench + SWE-smith + SWE-Gym | 0.807 | 0.067 | **0.612** | 0.766 | **0.033** | **0.718** |

Table 4: Comparison of different verifiers on three core abilities. Evaluation trajectories are sampled from `Qwen3-Coder` and `OpenHands-LM-32B` on SWE-bench Verified. EB means execution-based verifier while EF stands for execution-free verifier. Best results are in bold.

| VERIFIER | TYPE | OpenHands-LM-32B | | | Qwen3-Coder-Flash | | | Qwen3-Coder-Max | | |
|---|---|---|---|---|---|---|---|---|---|---|
| | | AUC | ECE↓ | RM@32 | AUC | ECE↓ | RM@32 | AUC | ECE↓ | RM@32 |
| AGENTLESS (Xia et al., 2024) | EB | - | - | 42.4% | - | - | 52.6% | - | - | 65.0% |
| SWE-GYM (Pan et al., 2025) | EF | 0.718 | 0.164 | 41.6% | 0.776 | 0.223 | 51.2% | 0.752 | 0.283 | 65.4% |
| DEEP SWE (Luo et al., 2025) | EB | - | - | 44.2% | - | - | 54.6% | - | - | 67.6% |
| | EF | 0.732 | 0.118 | 44.6% | 0.758 | 0.124 | 53.2% | 0.74 | 0.139 | 66.2% |
| SWE-RM-30A3B | EF | **0.748** | **0.080** | **48.8%** | **0.783** | **0.051** | **62.0%** | **0.768** | **0.047** | **74.6%** |

## 5.1 A NEW STATE-OF-THE-ART IN TTS

Based on the investigation in § 4, our final trained SWE-RM achieves state-of-the-art performance compared with previous works. We begin by discussing the baselines and evaluation setup, followed by an analysis of the SWE-RM results on TTS, AUC, and calibration.

**Baselines** We compare our trained execution-free verifier against several existing execution-free and execution-based verifiers: (1) *Agentless* (Xia et al., 2024), an execution-based method that generates reproduction tests for each trajectory and re-ranks them based on test results; (2) *SWE-Gym Verifier* (Pan et al., 2025), an execution-free verifier based on `Qwen2.5-32B` and trained on the SWE-Gym dataset; (3) *DeepSWE-EB Verifier* (Luo et al., 2025), the execution-based component of the current state-of-the-art DeepSWE Hybrid-TTS. This verifier extends the R2E-Gym execution-based verifier (Jain et al., 2025) and follows a similar mechanism to Agentless; (4) *DeepSWE-EF Verifier* (Luo et al., 2025), the execution-free component of DeepSWE Hybrid-TTS, which improves upon the R2E-Gym execution-free verifier. The evaluation setup for SWE-RM are same as the setting illustrated in § 4.1.

**SWE-RM performance** Our results in Table 4 show that SWE-RM consistently outperforms all baselines across AUC, ECE, and RM@32, achieving the best TTS, discrimination and calibration ability. The gains are not limited to `Qwen3-Coder` series models, where RM@32 improves pass@1 by 7-10 points, but also extend to `OpenHands-LM-32B`, where SWE-RM delivers the highest overall performance. This demonstrates the generalization ability of our verifier.

## 5.2 REINFORCEMENT LEARNING WITH EXECUTION-FREE FEEDBACK IN SWE

Different from reinforcement learning in math problems, which easily receive scalable, correct reward by comparing with the ground truth answers, reinforcement learning from verifiable reward (RLVR) are facing two major challenges in software engineering tasks: (1) Most training data are constructed by some automated pipelines with unchecked quality unit tests, the execution-based feedback are not guaranteed to be correct. (2) The long horizon context length and sandbox ex-

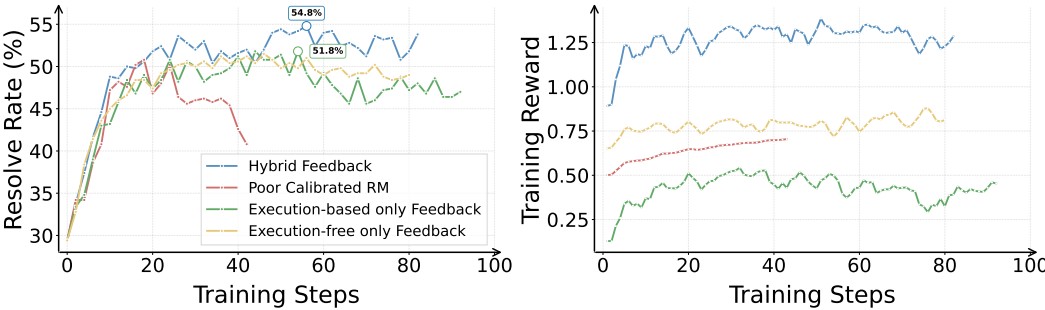

Figure 7: Left: RL performance on SWE-bench Verified when using different feedback. Right: Average training reward for different models.

ecution significantly limit the scale of RL, leading to slow improvements especially under sparse 0/1 reward. In this subsection, we show execution-free feedback offers a promising approach to not only accelerate training but also further enhance overall performance by providing more fine-grained reward signals. We first establish the RL setup in § 5.2.1 and then discuss the results in § 5.2.2.

### 5.2.1 RL Setup

**Scaffold and Model** For training scaffold, we adapt verl (Sheng et al., 2024) with Megatron (Shoeybi et al., 2019) which enables efficient multi-turn agentic reinforcement learning and SGLang for trajectory rollout. For agent scaffold, similar to the reward model rollout, we employ OpenHands (Wang et al., 2025) for tool interactions. For base models, we use `Qwen3-30B-A3B` (Qwen Team, 2025) with warm-up.

**Evaluation Setup** Similar to reward model evaluation, we conduct our evaluation on SWE-bench Verified (Jimenez et al., 2024). In the RL setting, we only generate 1 trajectory and 1 patch for each instance using greedy decoding following OpenHands (Wang et al., 2025), without any test-time scaling, as the final pass@1 score.

**Baselines** We compare different types of feedback during reinforcement learning: (1) *Hybrid feedback*, where the feedback is a combination of execution-free (SWE-RM) and execution-based signals, as will be defined in Eq. 1; (2) *Execution-free feedback only*, where the feedback is provided solely by SWE-RM; (3) *Execution-based feedback only*, where the feedback is derived exclusively from the execution results of fail2pass tests; (4) *Poorly calibrated execution-free feedback*, where the feedback comes from a reward model with comparable TTS but lower AUC and weaker calibration ability.

**Implementation Details** We adapt GSPO (Zheng et al., 2025) which provides greater stability for Mixture-of-Experts RL training and we define the execution-free feedback as $\mathrm{Score}_{EF}(q, \tau, \mathrm{patch}) \in [0, 1]$, and the overall reward is computed as:

$$r(q, \tau_i) = \begin{cases} 1 + \mathrm{Score}_{EF}(q, \tau_i, \mathrm{patch}_i), & \text{if issue resolve,} \\ -0.5 + \mathrm{Score}_{EF}(q, \tau_i, \mathrm{patch}_i), & \text{unfinished,} \\ 0 + \mathrm{Score}_{EF}(q, \tau_i, \mathrm{patch}_i), & \text{otherwise.} \end{cases} \tag{1}$$

More details about the RL training such as data, hyper-parameters are shown in Appendix E.

### 5.2.2 Execution-free Feedback benefits RL training

As shown in Figure 7, using the hybrid reward as described in Eq. 1 yields the best RL performance and efficiency. Compared to the execution-based baseline, hybrid feedback improves pass@1 by about 3 absolute points (54.8% vs. 51.8%) and shows faster, smoother improvements, indicating effective reward shaping. For execution-based feedback only, we observe slower early gains and an early plateau due to the sparsity of the 0/1 signals and issues with test noise and coverage. In

Table 5: Performance after RL on different SWE tasks other than SWE-Bench Verified, and Terminal Bench when using different feedback. SW.B. is short for SWE-Bench and Bold stands for the best.

| METHOD | SW.B. LIVE (LITE) | SW.B. MULTILINGUAL | MULTI-SW.B. MINI | TERMINAL BENCH |
|---|---|---|---|---|
| Hybrid | **22.4** | **35.7** | **20.0** | **32.5** |
| Execution-free only | 20.4 | 33.0 | 18.8 | 31.3 |
| Execution-based only | 20.0 | 33.3 | 18.5 | 30.0 |
| Poor Calibrated RM | 12.0 | 21.0 | 10.0 | 15.0 |

contrast, execution-free feedback alone shows faster initial progress due to continuous signals but weaker convergence in later stages, likely caused by inaccuracies in its unverified signals. While our main evaluation focuses on SWE-Bench Verified, we additionally conducted experiments on a broader suite of SWE tasks—including SWE-Bench Live (Lite), SWE-Bench Multilingual, Multi-SWE-Bench Mini, and Terminal Bench—to assess generalization beyond the original domain. As shown in Table 5, hybrid feedback consistently achieves better RL performance, while execution-free only feedback shows comparable results to execution-based only feedback. And if using feedback from a poorly calibrated RM, the model will also show a significant decrease in other tasks. Overall, combining execution-free feedback with verifiable signals balances efficiency and reliability, achieving the strongest final results by providing both continuous and trustworthy rewards.

## 6    CONCLUSION

In this paper, we show that test-time scaling alone is an insufficient measure of verifier quality for SWE agents. Beyond top-1 ranking, reward models must also deliver strong discrimination (AUC) and reliable calibration (low ECE) to provide stable, useful signals, especially for RL. Guided by large-scale ablations on data scale, positive/negative ratios, policy mixtures, source composition etc., we develop SWE-RM—a 30B MoE (3B activated) execution-free verifier with up to 256k context. SWE-RM achieves state-of-the-art open-source TTS gains on SWE-Bench Verified and, when used for RL, yields faster, more stable training and +3 absolute pass@1 over execution-based feedback counterparts. This establishes execution-free, well-calibrated reward modeling as a practical and powerful foundation for advancing SWE agents in both TTS and RL.

## 7    ACKNOWLEDGMENTS

This project is partially supported by Hong Kong RGC ECS Grant 26218125, Hong Kong RGC CRF Grant C6003-24Y, and NSFC Grant 62306177.

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

Table 6: Comparison of SWE-RM with other execution-free verifiers' setting: SWE-Gym Verifier (Pan et al., 2025), R2E-Gym Verifier (Jain et al., 2025), OpenHands Critic (Team, 2025) and DeepSWE Verifier (Luo et al., 2025). - means the statistic are not disclosed. *: Our training data source contains SWE-Gym (Pan et al., 2025), R2E-Gym (Jain et al., 2025), SWE-smith (Yang et al., 2025) and SWE-rebench (Badertdinov et al., 2025).

| REWARD MODEL | # DATA | # REPO | POLICY | SOURCE | CONTEXT LEN. | PROMPT |
|---|---|---|---|---|---|---|
| SWE-Gym Verifier | 2636 | 11 | Mix | SWE-Gym | 32k | Traj. |
| R2E-Gym Verifier | 3321 | 10 | Claude-3.5 | R2E-Gym | 32K | Traj. + Patch |
| Openhands Critic | - | 11 | - | SWE-Gym | 32k | Traj. |
| DeepSWE Verifier | - | 10 | - | R2E-Gym | 76k | Traj. + Patch |
| SWE-RM | $\sim$ 100k | $\sim$ 170 | Mix | Multiple* | 256k | Traj. + Patch |

## A  THE USE OF LARGE LANGUAGE MODELS

For this paper, large language models(LLMs) are used solely for polishing the writing. The entire research process, including but not limited to ideation, was conducted without any assistance from LLMs.

## B  EXTENDED RELATED WORK

### B.1  VERIFIERS FOR SWE TASKS

In Software Engineering (SWE) tasks, there are mainly two kinds of verifiers: (1) execution-based verifiers and (2) execution-free verifiers. Execution-based verifiers typically consist of various types of unit tests, either human-written or model-generated. Agentless (Xia et al., 2024), R2E-Gym (Jain et al., 2025), and DeepSWE (Luo et al., 2025) have demonstrated the effectiveness of execution-based verifiers in test-time scaling by ranking patches based on the number of unit tests passed. However, execution-based verifiers struggle to distinguish between patches that achieve the same number of test passes, and they suffer from the inherent unreliability of poorly written or model generated unit tests. In contrast, execution-free verifiers are typically model-based and provide a continuous score for a given trajectory, allowing finer-grained discrimination. Early work such as SWE-Gym (Pan et al., 2025) and OpenHands Critic (Team, 2025), has initially explored naive execution-free verifiers with limited coverage and were confined to relatively simple settings. Subsequent work, including R2E-Gym (Jain et al., 2025) and DeepSWE (Luo et al., 2025), has shown that combining execution-based and execution-free verifiers leads to improved test-time scaling. Nevertheless, as summarized in Table 6, the exploration of execution-free verifiers remains preliminary and has largely focused only on scaling performance. Different from them, this work first demonstrates that test-time scaling (TTS) performance alone is not a sufficient measure of an execution-free verifier's quality – its accuracy and calibration are equally important. We further present a systematic study on training versatile execution-free verifiers, considering factors such as data scaling, data ratio, policy mixture, source mixture, and context length.

### B.2  AGENTIC REINFORCEMENT LEARNING FEEDBACK IN SWE TASKS

Unlike non-agent scaffolds such as Agentless (Xia et al., 2024), which are single-turn and pipeline-based, agent scaffolds in SWE such as OpenHands (Wang et al., 2025) and SWE-Agent (Yang et al., 2024) deploy a sandbox environment that allows models to interact in a multi-turn setting. Upon completion, a fail-to-pass unit test is executed to assess whether the generated patch resolves the issue. This type of execution-based feedback plays a role similar to that of rule-based metrics in math problems (DeepSeek-AI, 2025), as it aims to provide a verifiable and relatively accurate reward for reinforcement learning. Such feedback has been widely adopted in recent coding agent training, including Qwen3-Coder (Qwen Team, 2025), GLM-4.5 (Team et al., 2025a), and MiniMax-M1 (MiniMax, 2025). While effective in principle, execution-based feedback is limited by the quality of the test suites it relies on. It assumes that the test oracle is perfect, yet in practice, many test cases are model-generated, making them an unreliable proxy for correctness. In addition, execution-based feedback cannot distinguish between trajectories that yield the same outcome, either passing or fail-

ing a test. Consequently, it often provides signals that are overly sparse or even misleading for reinforcement learning. In this work, we are the first to integrate execution-free feedback into SWE agentic reinforcement learning. We find that a versatile execution-free feedback can offer more fine-grained rewards, and lead to improved training efficiency and performance.

## C  THEORETICAL LINK BETWEEN TTS, AUC, AND ECE AND RL DYNAMICS

In this section, we make explicit how the three metrics—TTS, AUC, and ECE—correspond to three distinct failure modes of reward models (RMs) when used as optimization signals in RL. Throughout this section, let $\tau \sim \pi_\theta$ denote a sampled trajectory from the policy, $r(\tau) \in [0, 1]$ is the RM score, and $c(\tau) \in \{0, 1\}$ the binary correctness label (the ideal true correctness). If the RM score is used directly as the reward, the policy-gradient update is

$$\nabla_\theta J(\theta) \approx \mathbb{E}_{\tau \sim \pi_\theta}[r(\tau) \nabla_\theta \log \pi_\theta(\tau)]. \tag{2}$$

The ideal update using the true correctness label is

$$\nabla_\theta J^\star(\theta) = \mathbb{E}_{\tau \sim \pi_\theta}[c(\tau) \nabla_\theta \log \pi_\theta(\tau)]. \tag{3}$$

The gap between (2) and (3) determines the stability and correctness of RL. Here we use simple policy gradient as an example, which has same nature to GRPO (DeepSeek-AI, 2025)/GSPO (Zheng et al., 2025), one can simply extend the analysis to GRPO setting. We then analyze below how TTS, AUC, and ECE each correspond to a different component of this gap.

### C.1  TTS: EXTREME-TOP ERRORS AND THEIR IMPACT ON RL

TTS@k evaluates whether the highest-scored trajectory among $k$ RM-scored samples is correct:

$$\text{TTS@}k = \Pr(c(\tau^\star) = 1), \qquad \tau^\star = \arg \max_{i \in [1:k]} r(\tau_i). \tag{4}$$

Thus $(1 - \text{TTS@}k)$ is exactly the probability that an *unresolved* trajectory receives the *largest* reward among the $k$ samples.

**Implication for RL**  When the top-1 trajectory $\tau^\star$ is incorrect ($c(\tau^\star) = 0$), we know that

$$r(\tau_-^\star) = \max_{i \in [1:k]} r(\tau_i).$$

This does *not* imply that negatives have larger average rewards than positives, but it *does* imply that the batch contains a **negative trajectory with the largest reward weight**.
Since policy-gradient updates scale linearly with reward,

$$\nabla_\theta J(\theta) \approx \sum_{i=1}^{k} r(\tau_i) \nabla_\theta \log \pi_\theta(\tau_i),$$

the contribution of $\tau_-^\star$ becomes a *dominant* term in the update, even if all other negatives have small scores.
Conditioning on correctness of the top-ranked sample gives the decomposition:

$$\nabla_\theta J(\theta) = \text{TTS@}k \cdot \mathbb{E}[\text{update} \mid c(\tau^\star) = 1] + (1 - \text{TTS@}k) \cdot \mathbb{E}[\text{update dominated by } \tau_-^\star]. \tag{5}$$

Under the event $c(\tau^\star) = 0$, RL receives the strongest possible reward signal for an unresolved trajectory. This causes the policy $\pi_\theta$ to increase the probability of sampling this undesirable behavior, and the effect compounds over iterations.

### C.2  AUC: PAIRWISE RANKING QUALITY AND REVERSED-GRADIENT FREQUENCY

AUC measures the probability that the RM correctly orders a positive trajectory above a negative one (for all trajectories):

$$\text{AUC} = \Pr(r(\tau_+) > r(\tau_-)). \tag{6}$$

Therefore the mis-ranking probability is

$$1 - \text{AUC} = \Pr(r(\tau_-) > r(\tau_+)). \tag{7}$$

**Mis-rankings imply reversed gradient contributions** Whenever $r(\tau_-) > r(\tau_+)$, the RM assigns higher reward to an incorrect trajectory. The corresponding policy-gradient contributions satisfy

$$r(\tau_-)\,\nabla_\theta \log \pi_\theta(\tau_-) \;>\; r(\tau_+)\,\nabla_\theta \log \pi_\theta(\tau_+), \tag{8}$$

which is *opposite* to what the ideal update (3) would encourage. Thus a fraction $1 - \mathrm{AUC}$ of all positive–negative trajectory pairs induce updates pointing in the wrong direction. Conditioning on the correctness of ranking yields the decomposition

$$\nabla_\theta J(\theta) \;\approx\; \mathrm{AUC} \cdot \mathbb{E}[\text{correct pair update}] + (1 - \mathrm{AUC}) \cdot \mathbb{E}[\text{reversed pair update}]. \tag{9}$$

**Consequences for RL stability** Since gradient contributions aggregate linearly, the expected fraction of bad (reversed) updates grows exactly in proportion to $(1 - \mathrm{AUC})$. Therefore,

$$\text{low AUC} \implies \text{many reversed-gradient terms} \implies \text{unstable or divergent RL behavior.}$$

Unlike TTS (which concerns only the extreme top), AUC measures *global ranking correctness*, which affects *every* sampled trajectory in RL.

## C.3 ECE: Calibration Error and Systematic Bias in RL Updates

For a reward model that outputs a score $r(\tau)$ (interpreted as confidence in "this trajectory is good") and a binary "good/bad" ground truth $c \in \{0, 1\}$. A reward model is calibrated if

$$\Pr(c = 1 \mid r = \alpha) = \alpha, \quad \forall \alpha \in [0, 1]. \tag{10}$$

Using the binned approximation for the reward-model scores across a dataset of trajectories, we can compute:

$$\mathrm{ECE} = \sum_{m=1}^{M} \frac{|B_m|}{n} \left| \mathrm{acc}(B_m) - \mathrm{conf}(B_m) \right|. \tag{11}$$

where $B_m$ is the divided bins and:

$$\mathrm{conf}(B_m) = \frac{1}{|B_m|} \sum_{i \in B_m} r_i, \qquad \mathrm{acc}(B_m) = \frac{1}{|B_m|} \sum_{i \in B_m} c_i$$

**Calibration and unbiased RL updates** If Eq. (10) holds, then $\mathbb{E}[c \mid r] = r$, e.g. For all trajectories to which the model assigns a confidence score of ($r = 0.7$), the actual proportion of successful trajectories should also be 70%. This is exactly the statement ($\mathbb{E}[c \mid r = 0.7] = 0.7$), meaning it matches the model's own predicted confidence ($r$). And thus

$$\mathbb{E}[r(\tau)\nabla_\theta \log \pi_\theta(\tau)] = \mathbb{E}[c(\tau)\nabla_\theta \log \pi_\theta(\tau)], \tag{12}$$

meaning the RM induces *no systematic bias* in the expected gradient.

**Bias induced by miscalibration** In general, the deviation between the RM-induced and ideal updates is

$$\Delta_{\mathrm{bias}} = \mathbb{E}_{\tau \sim \pi_\theta}\left[ \nabla_\theta \log \pi_\theta(\tau)\left( r(\tau) - \mathbb{E}[c \mid r(\tau)] \right) \right]. \tag{13}$$

Define the calibration bias function

$$b(\alpha) = \mathbb{E}[c \mid r = \alpha] - \alpha,$$

so that

$$r(\tau) - \mathbb{E}[c \mid r(\tau)] = -b(r(\tau)).$$

Here $b(\alpha)$ measures the *calibration bias* at confidence level $\alpha$: among all trajectories for which the RM predicts score $r = \alpha$, $\mathbb{E}[c \mid r = \alpha]$ is the true success frequency, while $\alpha$ is the predicted success probability. Their difference therefore captures the systematic over- or under-confidence of the reward model at that score. ECE is then a binned approximation of the expected magnitude of this bias over the score distribution, meaning high ECE $\Rightarrow$ large systematic distortion in (13).

**Additional effect: gradient variance inflation.** Write the RM-induced gradient estimator as

$$g(\tau) = r(\tau) \, \nabla_\theta \log \pi_\theta(\tau).$$

Using the decomposition

$$r(\tau) = \mathbb{E}[c(\tau) \mid r(\tau)] - b\big(r(\tau)\big),$$

we obtain

$$g(\tau) = \underbrace{\mathbb{E}[c(\tau) \mid r(\tau)] \, \nabla_\theta \log \pi_\theta(\tau)}_{g^\star(\tau)} - \underbrace{b\big(r(\tau)\big) \, \nabla_\theta \log \pi_\theta(\tau)}_{\delta g(\tau)}.$$

Here $g^\star(\tau)$ is the "calibrated" part of the gradient (which would be obtained if we replaced $r(\tau)$ by the true success probability $\mathbb{E}[c(\tau) \mid r(\tau)]$), while $\delta g(\tau)$ is a purely micalibration-induced noise term. The variance of $g(\tau)$ decomposes as

$$\mathrm{Var}[g(\tau)] = \mathrm{Var}[g^\star(\tau)] + \mathrm{Var}[\delta g(\tau)] + 2 \, \mathrm{Cov}\big(g^\star(\tau), \delta g(\tau)\big).$$

In particular, we always have

$$\mathrm{Var}[g(\tau)] \;\geq\; \mathrm{Var}[g^\star(\tau)],$$

with the excess variance controlled by

$$\mathrm{Var}[\delta g(\tau)] = \mathrm{Var}\big[b(r(\tau)) \, \nabla_\theta \log \pi_\theta(\tau)\big].$$

Thus calibration error $b(r)$ couples multiplicatively with the policy gradient $\nabla_\theta \log \pi_\theta(\tau)$, injecting additional variance into the gradient estimator. Since ECE is a discrete approximation of $\mathbb{E}_r\big[\,|b(r)|\,\big]$, higher ECE typically implies a larger variance contribution from $\delta g(\tau)$ and therefore less stable RL training.

## D  REWARD MODEL TRAINING DETAILS

### D.1  DETAILED TRAINING SETUP

**Data Collection**  To train a reward model, we first rollout and collect over 400k multi-turn trajectories up to 100 iterations using OpenHands (Wang et al., 2025) and SWE-Agent (Yang et al., 2024), which are two widely used open-sourced coding agent scaffold using different policy models and data sources, including SWE-Gym (Pan et al., 2025), SWE-rebench (Badertdinov et al., 2025), SWE-smith (Yang et al., 2025), and R2E-Gym (Jain et al., 2025). Specifically, we adapt `Qwen3-Coder-Max`, `Qwen3-Coder-Flash`, `Claude-4-sonnet` for rollout. Since a large portion of the data might be unresolved and some trajectories may be incomplete or contains bad tool calls, thus the final usable trajectories for training is around 100k.

These trajectories are then labeled as positive(resolved) or negative(unresolved) based on their execution results with the provided fail2pass test. Though some of them might be noisy as we discussed that unit tests might not be able to truly reflect the correctness of the generated patch, we applied data cleaning such as filtering out instances without any successful trajectory (typically cases affected by over-strict/unfair unit tests or under-specified descriptions) to maintain the highest possible label quality. With data filtering, we believe a sufficiently large and diverse dataset enables the RM to learn a denoised and generalized correctness signal despite noisy supervision.

**Scaffold and Model**  For training scaffold, we adapt Megatron (Shoeybi et al., 2019) for supervised fine-tuning which enables efficient long context training. For rollout, we use SGLang together with agent scaffold OpenHands (Team, 2025) and SWE-Agent (Yang et al., 2024). While for the base model, we use `Qwen3-30B-A3B` (Qwen Team, 2025) as the backbone for further training. This MoE architecture is not a choice that claiming the calibration advantages,rather, we follow the prevailing practice in state-of-the-art coding agents (e.g., Qwen3-Coder, Kimi-K2, MiniMax-M1/2), which predominantly adopt MoE backbones. Using the same backbone ensures compatibility with existing pipelines(e.g. infra) and allows our reward model and trained policies to be directly integrated without additional adaptation overhead. Our focus is therefore on reward-model training and calibration, while architectural comparisons (MoE vs. Dense vs. Adapters) are left to future work.

**Baselines**  We compare our trained execution-free verifier with the following different execution-free verifiers as well as execution-based verifiers: (1) *Agentless* (Xia et al., 2024), which proposes an execution-based method that generates reproduction tests for each trajectory and re-ranks based on passed test numbers; (2) *SWE-Gym Verifier* (Pan et al., 2025), which releases a `Qwen2.5-32B` based execution-free verifier trained on SWE-Gym; (3) *DeepSWE-EB Verifier* (Luo et al., 2025): which is the execution-based component of current State-of-the-art DeepSWE Hybrid-TTS, which is also the improved version of R2E-Gym Execution-based verifier (Jain et al., 2025) with similar mechanism to Agentless. (4) *DeepSWE EF Verifier* (Luo et al., 2025): which is the execution-free component of current State-of-the-art DeepSWE Hybrid-TTS also the improved version of R2E-Gym execution-free verifier.

**Evaluation Setup**  We mainly conduct the evaluation on SWE-bench Verified (Jimenez et al., 2024) which is a curated subset of 500 human-verified tasks for reliably assessing model performance on real-world software engineering tasks. For test-time scaling and further accuracy and calibration evaluation, we use the most widely used open-sourced coding agent scaffold Open-Hands to collect 32 independent runs for each instance, resulting $32 \times 500$ trajectories in total. The sampling configs use a temperature of 1.0, top_p of 0.95, max_iterations of 100. Accuracy is calculated by AUC score on all 16k trajectories while calibration is measured by expected calibration error(Guo et al., 2017), where $\text{ECE} = \sum_{m=1}^{M} \frac{|B_m|}{n} \Big| \operatorname{acc}(B_m) - \operatorname{conf}(B_m) \Big|, \operatorname{conf}(B_m) = \frac{1}{|B_m|} \sum_{i \in B_m} r_i, \ \operatorname{acc}(B_m) = \frac{1}{|B_m|} \sum_{i \in B_m} c_i$. And we follow common practice to divide confidence into 10 bins ($M = 10$).

**Pass**@k defines the resolve rate of model with at least one successful solution among k trajectories which is also the upper bound while **RM**@K defines the resolve rate of final selected trajectories from k samples. For every k < 32, we obtain the mean and variance of 5 random runs for fair assessment.

## D.2 TRAINING TEMPLATE

Following SWE-Gym (Pan et al., 2025), we adapt from their template which splices all turns (model action output and tool responses) and end with a YES/NO token for reward model classification. Given the full multi-turn trajectory, the model is prompted to output a single special token, either <YES> (resolved) or <NO> (unresolved). And the supervised fine-tuning utilizes standard next-token prediction loss on this special token. At inference time, by obtaining the log probability of the special token <YES>($l_y$) and <NO>($l_n$), the final score $r$ is calculated by $\exp(l_y)/(\exp(l_y) + \exp(l_n))$, which maps to a continuous reward model score $r \in [0, 1]$. For tool parsing, we adapt `Qwen3-Coder`'s XML format, which optimized for code-related argument parsing. The example trajectory is shown in Figure 8.

## D.3 TRAINING HYPER-PARAMETERS

For reward model training hyperparameters, we adapt a 256k context window to support scoring for complex questions which contains extremely long contexts. The global batch size is set to 128. Also widely used AdamW optimizer and cosine decay learning rate scheduler are used. The detailed training hyperparameters are listed in Table 7 below. We use 4 nodes of H100 for large scale, long context reward model training, which takes around 20 hours for 100k samples.

Table 7: The detailed training hyperparameters for reward model.

| Model Size | Global Batch Size | Learning Rate | Schedular | LR Warmup | Optimizer | Epoch |
|---|---|---|---|---|---|---|
| 30BA3 | 128 | 7e-6 → 7e-7 | cosine | 3% | AdamW | 1 |
| 235BA22 | 128 | 7e-6 → 7e-7 | cosine | 3% | AdamW | 1 |
| 480BA35 | 128 | 7e-6 → 7e-7 | cosine | 3% | AdamW | 1 |

We also summarize the detailed training setting for Verifiers we used (Verifier A, Verifier B and Poor calibrated RM in Figure 7) in Table 8 below.

```
<user>
Please evaluate the following interaction between an AI assistant and a user:
=== INTERACTION LOG ===
 *** System Message that describes the assistant's behavior ***
[system message here]
------------------------------------------------------------------
*** Turn 1 - USER ***
...
------------------------------------------------------------------
*** Turn 1 - ASSISTANT ***
...
------------------------------------------------------------------
...
------------------------------------------------------------------
*** Turn 9 - ASSISTANT ***
Let's ignore the existing error and focus on our specific issue. ...
<tool_call>
<function=str_replace_editor>
<parameter=path>
/workspace/astropy__specreduce__1.2/specreduce/tests
</parameter>
<parameter=command>
view
</parameter>
</function>
</tool_call>
------------------------------------------------------------------
*** Turn 10 - USER ***
<tool_response>
Here's the files and directories up to 2 levels deep in /workspace/
astropy__specreduce__1.2/specreduce/tests,
...
</tool_response>
------------------------------------------------------------------
...
=== END INTERACTION ===
=== GENERATED PATCH ===
[patch here]
=== END GENERATED PATCH ===
Based on the above interaction and the generated patch, did the assistant
successfully resolve the user's initial request? Respond with YES or NO.
</user>
<assistant>
<judgement>NO
```

Figure 8: Prompt template for reward model training. Pink refer to the tool parsing XML format example.

## D.4    ADDITIONAL ANALYSIS ON CONTEXT CONSTRAINT

In § 4.3 we show a substantial performance increase when we raise the context window from 32k to 256k tokens. The larger context allows the model to cover far more tokens of the input (for

Table 8: Comparison of detailed training setting of Verifier A, Verifier B in Figure 2,3,4,5 and Poorly Calibrated RM in Figure 7.

| REWARD MODEL | # DATA | RATIO | POLICY | SOURCE | CONTEXT LEN. |
|---|---|---|---|---|---|
| Verifier A | 20k | 2:1 | Mix-Policy | Mixed-Source | 256k |
| Verifier B | 20k | 1:4 | Off-Policy | SWE-Rebench | 256k |
| Poorly Cali. RM in Figure 7 | 5k | 1:2 | Off-Policy | SWE-Rebench | 256k |

example, longer trajectories, multi-file code, richer history) without needing to compress or truncate information. When the window is small, many long trajectories cannot be scored at all (i.e., fall out of the window) and thus receive no valid reward signal, meaning correct but long solutions are effectively dropped from TTS selection. With a 256 k context, we reduce the "no-score" rate (increase the score rate), thereby enabling full-trajectory scoring and enabling our verifier to pick up solutions that would otherwise be ignored. And another reason for us to insist on 256k reward model training is that modern high-end coding agent policy models such as Claude, GPT-5, Qwen3-Coder-Max support 256k token windows (and up to 1 M tokens in some settings). The community therefore urgently needs a reward model that is compatible with this scale, so trajectories from such long-context agents can be properly scored — yet many existing reward/verifier models are constrained to much shorter context windows and thus cannot handle those long trajectories.

By aligning our verifier to the 256 k-token scale we fill a crucial gap. Finally, we acknowledge that increasing the context to 256k tokens is not free — it incurs higher memory usage. However, since our output generation only contains one token, all prompt computation can be run in parallel, thus the latency is more or less the same for different context length at inference time. While for deployment, a 256k model takes only around 2x GPU memory usage than a 32k model, a user with 2x A100 GPUs can easily deploy the model with high efficiency.

# E  RL TRAINING DETAILS

## E.1  RL SETUP

**Models**   We conduct reinforcement learning based on a `Qwen3-30B-A3B` (Qwen Team, 2025) with SFT warm-up. WE use in-house collected agentic trajectories including but not limited to SWE tasks to fine-tune the base model. Then this fine-tuned model served as the starting point of our RL experiments. We use MoE architecture, following the prevailing practice in state-of-the-art coding agents (e.g., Qwen3-Coder (Qwen Team, 2025), Kimi-K2 (Team et al., 2025b), MiniMax-M1/2 (MiniMax, 2025)), which predominantly adopt MoE backbones. Using the same backbone ensures compatibility with existing pipelines(e.g. infra) and allows our trained policies to be directly integrated without additional adaptation overhead.

**Implementation Details**   We train the model using curated data from SWE-Gym (Pan et al., 2025) and SWE-rebench (Badertdinov et al., 2025) with a batch size of 64. For each problem, we sample 16 rollouts, use a maximum of 100 iterations, and set the context length to 128k. This context length was selected because it accommodates most problem cases within a single context window while being more cost-efficient than a 256k context window. The training data is further filtered by difficulty, as problems that are either too easy or too difficult can negatively affect RL performance.

Unlike single-turn RL training, where the model needs to generate only one response per problem, agentic RL requires interaction with an agent scaffold (i.e., tool calls) across multiple turns to construct a full trajectory. Specifically, given a problem $q$, the policy model generates an action $a_i$ and then receives a tool response $o_i$, repeating this process $T$ times to form a trajectory $\tau = \{a_1, o_1, a_2, o_2, \ldots, a_T, o_T\}$. During optimization, tool responses are masked. Instead of the GRPO objective, we adopt the GSPO objective (Zheng et al., 2025), which provides greater stability, particularly in Mixture-of-Experts (MoE) RL training:

$$\mathcal{J}_{\text{GSPO}}(\theta) = \mathbb{E}_{q \sim \mathcal{D}, \{\tau_i\}_{i=1}^{G} \sim \pi_{\text{old}}(\cdot | q)}$$

$$\left[ \frac{1}{G} \sum_{i=1}^{G} \min \left( \left( \frac{\pi_\theta(\tau_i \mid q)}{\pi_{\theta_{\text{old}}}(\tau_i \mid q)} \right)^{\frac{1}{|\tau_i|}} \widehat{A}_i, \text{clip} \left( \left( \frac{\pi_\theta(\tau_i \mid q)}{\pi_{\theta_{\text{old}}}(\tau_i \mid q)} \right)^{\frac{1}{|\tau_i|}}, 1 - \varepsilon, 1 + \varepsilon \right) \widehat{A}_i \right) \right] \quad (14)$$

with the group-based advantage estimation:

$$\widehat{A}_i = \frac{r(q, \tau_i) - \text{mean}\left(\{r(q, \tau_i)\}_{i=1}^G\right)}{\text{std}\left(\{r(q, \tau_i)\}_{i=1}^G\right)} \tag{15}$$

The optimization integrate several standard tricks such as Clip High (Yu et al., 2025), NO KL loss (Yu et al., 2025) etc.

