# OpenReview forum: "SWE-RM: Execution-free Feedback for Software Engineering Agents"
_ICLR.cc/2026/Conference — ICLR 2026 Poster_

### Official Review · Reviewer_nzG1 · 2025-10-25

**Soundness:** 2
**Presentation:** 2
**Contribution:** 2
**Rating:** 4
**Confidence:** 3

**Summary:**

The paper presents SWE-RM, an execution-free reward model for software engineering agents that replaces unit-test feedback with model-based scoring. By introducing AUC and calibration metrics beyond test-time scaling, it achieves SOTA results and improves RL performance on SWE-Bench Verified.

**Strengths:**

1.	The paper addresses an important and emerging challenge in LLM based software engineering that how to design effective reward signals for coding agents when execution-based feedback is unreliable or unavailable.
2.	The authors provide a convincing empirical motivation showing that similar TTS scores can lead to drastically different RL outcomes.
3.	Section 4 includes a broad set of ablations on training data scale, data ratio, policy mixture, and source composition. These controlled studies give a practical recipe for training more robust reward models.
4.	The proposed SWE-RM achieves consistent performance gains on SWE-Bench Verified, improving both test-time scaling results and downstream RL outcomes.

**Weaknesses:**

1.	The paper proposes that three metrics TTS, AUC, and ECE jointly define a “versatile” reward model, but offers no theoretical justification or analytical link to reinforcement learning dynamics. The framework remains purely empirical, leaving the causal relationship between these metrics and RL stability unexplained.
2.	The model adopts a large 30B-parameter MoE with 3B active experts, but the rationale behind this choice is missing. There is no ablation comparing MoE against dense or adapter-based architectures, nor any argument explaining why modularization benefits calibration or discrimination.
3.	The paper vaguely describes the reward model as predicting a special token (“YES/NO”) and mapping it to a continuous score. However, it never specifies the precise loss function. This omission makes it impossible to reproduce or understand the optimization process.
4.	Although the method claims to produce execution-free feedback, the training labels are derived from fail2pass execution tests. This dependency contradicts the notion of being execution-free and blurs the boundary between model-based and execution-based supervision.
5.	Section 4.3 shows performance improvements when increasing context length from 32k to 256k tokens, but provides no explanation of why this happens. The analysis ignores computational costs, latency trade-offs, and whether the gain comes from longer reasoning chains or merely from covering more tokens.
6.	SWE-RM employs a Mixture-of-Experts (30B total, 3B active) architecture, but the authors do not compare it with alternative designs. Nor do they explain why this structure improves reward calibration. The absence of such analysis weakens the methodological contribution.
7.	The paper simultaneously uses TTS, AUC, and ECE as evaluation metrics but never establishes their quantitative relationships or complementarity. For example, AUC and TTS both measure ranking ability; it is unclear why optimizing all three leads to better RL stability. The metric design does not appear to be theoretically motivated.
8.	The authors acknowledge that some trajectory labels are derived from fail2pass test results that may not reflect true correctness, yet they do not provide any estimate of label noise, data cleaning strategy, or robustness analysis.
9.	Despite claiming a 30B-parameter model trained on 100k samples, the paper omits key details such as GPU count, training duration, or energy cost. The reproducibility statement is overly brief and fails to include crucial hyperparameters, limiting verifiability.

**Questions:**

Please address the weaknesses noted above.

---

> ### Author Response · Authors · 2025-11-23
> **Rebuttal to Reviewer nzG1 [1]**
>
> Dear Reviewer nzG1,
> Thank you for your insightful comments! We truly appreciate that you found our work to provide convincing empirical motivation and consistent performance gains. We also noticed that a few aspects of our presentation may have led to some misunderstandings, and we clarify them point-by-point below:
>
> > The paper proposes that three metrics TTS, AUC, and ECE jointly define a “versatile” reward model, but offers no theoretical justification or analytical link to reinforcement learning dynamics. The framework remains purely empirical, leaving the causal relationship between these metrics and RL stability unexplained.
>
> Thanks for the comment. As we discussed in §3.2 via an illustrative example (Figure 3,4&5), TTS, AUC and ECE are all matters for RL, but let us now explicitly unpack how the three metrics connect to the RL reward‑model assumption.
> We also outlined a detailed theoretical analysis in Appendix C in updated paper. We would truly appreciate it if you could take a look at them. To summarize:
>
> (1) TTS: It is trivial that the TTS performance is directly related to RL signal quality. Because if the reward model has poor TTS performance, that means it tends to score unresolved trajectories as the highest, which implies the batch contains many negative trajectories with the largest reward weight. Detailed formula/proof can be checked in Appendix C.1.
>
> (2)AUC:  It quantifies the probability that the reward model correctly orders a positive trajectory above a negative one (for all trajectories), which is $\Pr\bigl(r(\tau_{+}) > r(\tau_{-})\bigr)$. Thus a fraction $1-\mathrm{AUC}$ of all positive--negative trajectory pairs induce updates pointing in the wrong direction, which destabilizes RL training. Unlike TTS (which concerns only the extreme top), AUC measures global ranking correctness, which affects every sampled trajectory in RL. Detailed formula/proof can be checked in Appendix C.2.
>
> (3) ECE: For a reward model that outputs a score $r$ (interpreted as confidence in “this trajectory is good”) and we observe a binary “good/bad” ground truth $c \in \\{0,1\\}$ , calibration demands:
> $\Pr(c = 1 \big| r = \alpha) = \alpha \quad\forall \alpha \in [0,1]$
>
> Using the binned approximation for the reward‑model scores across a dataset of trajectories, we can compute:
>
> $$\mathrm{ECE} = \sum_{m=1}^{M} \frac{|B_m|}{n}\Bigl|\mathrm{acc}(B_m) - \mathrm{conf}(B_m)\Bigr|,
> \mathrm{conf}(B_m) = \frac{1}{|B_m|}\sum_{i \in B_m} r_i,
> \mathrm{acc}(B_m) = \frac{1}{|B_m|}\sum_{i \in B_m} c_i$$
>
> If model is calibrated, then $\Pr(c = 1 \big| r = \alpha) = \alpha $holds, then $\mathbb{E}[c\mid r]=r$, e.g. For all trajectories to which the model assigns a confidence score of ($r$ = 0.7), the actual proportion of successful trajectories should also be 70\%. This is exactly the statement ($\mathbb{E}[c \mid r = 0.7] = 0.7$), meaning it matches the model’s own predicted confidence ($r$).
>
> Given the policy gradient update of the model is  $\nabla_\theta J(\theta) \approx \mathop{\mathbb{E}}_{\tau \sim \pi\_\theta} \[ r(\tau) \nabla\_\theta \log \pi\_\theta(\tau) \]$, we would obtain
> $\mathbb{E} [r(\tau) \nabla\_\theta\log\pi\_\theta(\tau)]=\mathbb{E}[c(\tau)\nabla\_\theta\log\pi\_\theta(\tau)]$ under a well-calibrated setting.
>
> This means the **RM induces no  bias in the expected gradient when it is perfectly calibrated**.
>
> We then further discuss the bias induced by mis-calibration. In general, the deviation between the RM-induced and ideal updates is
> $\Delta_{\mathrm{bias}}=\mathbb{E}_{\tau\sim\pi\_\theta}[\nabla\_\theta\log\pi\_\theta(\tau)\bigl(r(\tau)-\mathbb{E}[c | r(\tau)]\bigr)]$.
>
> Define the calibration bias function $b(\alpha)=\mathbb{E}[c\mid r=\alpha]-\alpha$, so that $r(\tau)-\mathbb{E}[c\mid r(\tau)]
> = -b(r(\tau))$.
>
> Here $b(\alpha)$ measures the calibration bias at confidence level $\alpha$: among all trajectories for which the RM predicts score $r = \alpha$, $\mathbb{E}[c\mid r=\alpha]$ is the true success frequency, while $\alpha$ is the predicted success probability.
>
> Their difference therefore captures the systematic over-or under-confidence of the reward model at that score. ECE is then a binned approximation of the expected magnitude of this bias over the score distribution, meaning high ECE $\Rightarrow$ large systematic distortion in the bias term. This is a summarized version and a more detailed formula/proof can be checked in Appendix C.3.

---

> ### Author Response · Authors · 2025-11-23
> **Rebuttal to Reviewer nzG1 [2]**
>
> > The model adopts a large 30B-parameter MoE with 3B active experts, but the rationale behind this choice is missing. There is no ablation comparing MoE against dense or adapter-based architectures, nor any argument explaining why modularization benefits calibration or discrimination.
>
> We would like to clarify that the 30B-parameter Mixture-of-Experts (MoE) backbone with 3B active parameters is not presented as a design choice in our paper, nor do we claim that MoE inherently improves calibration or discrimination.
>
> Our decision to build SWE-RM/test policy improvements based on the Qwen3-30B-A3B MoE simply follows the current best practice in coding agent models: the top-performing open-source coding agents (e.g., Qwen3-Coder, Kimi-k2, MiniMax-M1/2, etc.) are all MoE-based. Using the same architecture ensures that our RM and trained policy models are compatible with the strongest existing policies and allows the community to directly plug SWE-RM into prevailing training pipelines without extra adaptation overhead. And with limited computation budget, Qwen3-30B-A3B is the strongest open-source MoE coding model that our compute can afford.
>
> Because the focus of this work is how to train an accurate and well-calibrated reward model for TTS and RL, rather than which backbone architecture is optimal, we kept the backbone fixed and devoted our ablation budget to performance improvements. A systematic comparison of MoE vs. Dense vs. Adapter backbones for reward modeling/Policy is an interesting orthogonal question that we leave for future work.
>
> We have revised Appendix D.1 (Line 910 - 917) and Appendix E.1 (Line 1058 - 1062) to explicitly state that the MoE backbone is inherited from prior work and that our contribution not lies in architectural innovation.
>
> > The paper vaguely describes the reward model as predicting a special token (“YES/NO”) and mapping it to a continuous score. However, it never specifies the precise loss function. This omission makes it impossible to reproduce or understand the optimization process.
>
> Thank you for pointing out the need for clearer training details. We model reward model training as a generative classification task. Let  y $\in$ {YES, NO} denote the label for a given multi-turn trajectory. The model is trained with the standard next-token prediction (NTP) loss:
> $ \mathcal{L} = -\log P(y \mid \text{trajectory})$using the prompt template in Figure 8.
> At inference, the continuous reward score  r $\in$ [0,1]  is computed via:
> $ r = \frac{\exp(\ell_y)}{\exp(\ell_y) + \exp(\ell_n)},$
> where $\ell_y$ and $\ell_n$ are the log-probabilities of predicted tokens YES and NO, respectively.
>
> We have incorporated a clearer description of loss function and how we train the reward model and score calculation in §3.1 (Line 157-160) and §4.1 (Line 256 - 261). And we hope this could resolve your concern about reproducibility and understanding issues about the optimization process.
>
> > Although the method claims to produce execution-free feedback, the training labels are derived from fail2pass execution tests. This dependency contradicts the notion of being execution-free and blurs the boundary between model-based and execution-based supervision.
>
> Thank you for raising the point regarding our claim of “execution-free feedback”. We hereby clarify our position: the term “execution-free verifier” (or “model-based / execution-free feedback”) has indeed been defined and used in prior work in software-engineering (SWE) agent settings (for example, in that prior setting, the training labels for the execution-free verifier are often derived from execution-based outcomes (e.g., fail2pass tests) but at inference or deployment time the verifier does not perform environment execution—it is model-based and computes scores without running the tests. [1, 2] Our approach follows this exact paradigm: we use execution-based fail2pass tests to collect labeled trajectories, and then train a model (the verifier) that at inference/training time does not require executing the environment/tests to score solutions. We hope this clarification resolves the apparent contradiction.
>
> [1] Training Software Engineering Agents and Verifiers with SWE-Gym, J. Pan et al. ICML 2025.
>
> [2] R2E-Gym: Procedural Environments and Hybrid Verifiers for Scaling Open-Weights SWE Agents, N. Jain et al. COLM 2025

---

> ### Author Response · Authors · 2025-11-23
> **Rebuttal to Reviewer nzG1 [3]**
>
> > Section 4.3 shows performance improvements when increasing context length from 32k to 256k tokens, but provides no explanation of why this happens. The analysis ignores computational costs, latency trade-offs, and whether the gain comes from longer reasoning chains or merely from covering more tokens.
>
> First, on why: moving to a 256k token context enables two major advantages in our setting — (a) it covers much more of the input space, so that long trajectories, multi-file codebases, or extended reasoning chains are fully captured rather than truncated (in some settings); and (b) it grants the model with improved token-coverage: when the context length is shorter, many long trajectories cannot be scored at all (i.e., fall out of the window) and thus receive no valid reward signal, which biases the selection away from correct but long solutions. By contrast, with 256 k tokens the verifier can ingest and score all trajectories, reducing the “no-score” rate and enabling better test-time scaling and on-policy RL training.
>
> Second, on trade-offs: we recognise that increasing the context to 256 k tokens is not free — it incurs higher memory usage. However, since our output generation only contains one token, all prompt forwarding can be run in parallel, thus the latency is roughly the same for different context length at inference time, which is not a great concern as even 256k settings can be finished within one second. At our deployment, a 256k model takes only around 2x GPU memory usage than a 32k model, a user with 2x A100 GPUs can easily deploy the model with high efficiency. For more computation cost analysis of model training, we have included it in Q9 below. We have added the above analysis of context window in Appendix D.4 (Line 1023 - 1050).
>
> > SWE-RM employs a Mixture-of-Experts (30B total, 3B active) architecture, but the authors do not compare it with alternative designs. Nor do they explain why this structure improves reward calibration. The absence of such analysis weakens the methodological contribution.
>
> We believe this question overlaps with Question 2. For a detailed explanation, we would appreciate it if you could refer to our response to Q2.
>
> > The paper simultaneously uses TTS, AUC, and ECE as evaluation metrics but never establishes their quantitative relationships or complementarity. For example, AUC and TTS both measure ranking ability; it is unclear why optimizing all three leads to better RL stability. The metric design does not appear to be theoretically motivated.
>
> For the relationship between the three metrics and RL stability, we have added a comprehensive theoretical analysis in Q1 above, below we clarify the relationship and differences of these metrics.
>
> TTS, AUC, and ECE measure different aspects of verifier quality:
>
> First, TTS and AUC capture **distinct ranking behaviors**.
> - TTS focuses **only on top-1 ranking**: whether the correct trajectory is ranked highest among k samples. It is blind to the rest of the score distribution. Two verifiers can have identical TTS yet produce very different orderings for the remaining trajectories (as shown in Figure 5).
> - AUC, in contrast, evaluates **global discriminative ability** by integrating across all pairwise orderings of resolved vs. unresolved trajectories. A model can have high TTS but low AUC if it gets the top-1 decision correct while misordering most other near-miss trajectories. Such global misordering directly affects RL reward quality because RL repeatedly samples different trajectories, so the entire score landscape matters.
>
>
> Second, ECE provides information neither TTS nor AUC can capture.
> While TTS/AUC describe ranking, ECE measures **confidence and accuracy matching**, i.e., whether the score magnitude reflects the empirical likelihood of correctness. In RL, the reward is treated as an expectation proxy; when a verifier is miscalibrated (e.g., assigning 0.9 to trajectories that succeed only 50% of the time), policy gradients become biased, leading to unstable or collapsed updates (as we shown in Q1). This effect cannot be detected from ranking metrics alone.
> Thus, the three metrics are complementary and each of them contributes to the stability of RL dynamics. And we have strengthened the theoretical analysis in Appendix C. We hope these can resolve your concerns.

---

> ### Author Response · Authors · 2025-11-23
> **Rebuttal to Reviewer nzG1 [4]**
>
> > The authors acknowledge that some trajectory labels are derived from fail2pass test results that may not reflect true correctness, yet they do not provide any estimate of label noise, data cleaning strategy, or robustness analysis.
>
> Execution-based labels derived from fail2pass tests do contain noise. Following the analyses reported in OpenAI’s SWE-Bench annotation and subsequent studies, unit tests exhibit substantial noise:  28.3% of unit tests incorrectly flag valid solutions as failures (false negatives)[3], while [4] reports roughly 11% unit test verified solutions are indeed incorrect( false positives). This statistic could be an approximation to the noise existing in current SWE agent training data, which provides a reasonable estimate of the underlying label noise we inherit.
>
> To mitigate this, we apply basic data cleaning by filtering out instances with no successful trajectory (often caused by over-strict/unfair unit tests or under-specified descriptions). In addition, we rely on a large, diverse multi-policy multi-source dataset, which allows the reward model to statistically learn a denoised correctness signal despite noisy supervision. Empirically, this denoising effect is validated on the human-verified SWE-Bench Verified benchmark: SWE-RM reduces error rates to ~8% false positives and ~19.4% false negatives—substantially lower than the noise levels of the original execution labels.
>
> [3] Introducing SWE-bench Verified, OpenAI, Aug 13, 2024, https://openai.com/index/introducing-swe-bench-verified/
>
> [4] Are “Solved Issues” in SWE-bench Really Solved Correctly? An Empirical Study, You W. et,al. 2025
>
> > Despite claiming a 30B-parameter model trained on 100k samples, the paper omits key details such as GPU count, training duration, or energy cost. The reproducibility statement is overly brief and fails to include crucial hyperparameters, limiting verifiability.
>
> We have included all hyperparameters (optimizer, learning rate, batch size, training data sources, epoch etc.), implementation details and the exact training framework and scaffold (Megatron-LM, VeRL, openhands) in Appendix C.3(original pdf), and our implementation follows the training template which explicitly stated in Appendix C.2 (original pdf).
> We further discuss details that you are concerned about: we use 4 nodes of H100 for our main  reward model training, which takes around 20 hours for 100k samples. And we have added this information in Appendix D.3 (Line 960 - 962). With these details, the full optimization pipeline can be reproduced end-to-end. We hope this could resolve your concern about reproducibility.

---

### Official Review · Reviewer_aGbU · 2025-10-30

**Soundness:** 3
**Presentation:** 2
**Contribution:** 2
**Rating:** 2
**Confidence:** 3

**Summary:**

This paper proposes SWE-RM, an execution-free reward model for SE agents that provides continuous feedback without requiring unit test execution. The authors argue that execution-based feedback has limitations including sparsity and dependence on test quality, and propose using a learned reward model instead. They identify three key metrics for evaluating reward models: test-time scaling (TTS) performance, classification accuracy (AUC), and calibration (ECE). Through empirical ablations on training data scale, positive/negative ratios, policy mixtures, and data sources, they develop a 30B parameter mixture-of-experts model that achieves state-of-the-art results on SWE-Bench Verified for both test-time scaling and reinforcement learning.

**Strengths:**

The paper makes a reasonable observation that test-time scaling performance alone is insufficient for evaluating reward models intended for RL use. The empirical finding in Figure 2 showing that two verifiers with similar TTS can have drastically different RL performance is interesting and motivates the need for better evaluation criteria.
The reported improvements are good - lifting Qwen3-Coder-Flash and Max on SWE-Bench Verified represents meaningful progress.
Scaling the reward model to 256k context length addresses a practical limitation of prior work and enables scoring of complex, long trajectories.

**Weaknesses:**

The paper does not adequately address a critical limitation: verifying program correctness can be as difficult as, or even harder than, executing the program. A reward model must essentially predict execution outcomes across potentially many different code paths and edge cases without actually running the code. This requires the model to simulate program semantics, which is fundamentally challenging and may introduce systematic errors. The paper does not discuss: How the reward model handles corner cases or edge conditions that unit tests would catch. The types of errors the reward model systematically misses (false negatives) or incorrectly flags (false positives). Whether execution-free verification can ever be truly reliable for complex software engineering tasks. Why we should expect a learned model to generalize to unseen code patterns and bugs.
This represents a significant gap in the theoretical justification for the entire approach. The claim that execution-free feedback is superior needs much stronger grounding.

While the paper presents extensive ablations, the intuitions behind many design decisions are unclear:
Why is a 2:1 positive/negative ratio optimal? Is this simply an artifact of data availability, or is there a principled reason?
Why do mixed policies help? What specific distribution shift does this address?
The paper states "we believe reward model can still achieve relative good performance by training on a large number of data" (Appendix C.1) when discussing noisy labels, but provides no theoretical or empirical justification for this belief
Why are AUC and ECE the right metrics? While intuitively reasonable, there's no formal argument for why these metrics should predict RL performance

The core contribution appears to be training a larger reward model on more diverse data and identifying AUC/calibration as important metrics. The approach of training a classifier on successful/unsuccessful trajectories and using it for ranking is standard. The paper does not introduce novel algorithms, architectures (beyond using MoE), or training objectives. The identification of additional evaluation metrics, while useful, is incremental rather than constituting a significant methodological advance.

The reward model is trained on trajectories labeled by execution-based feedback, then claimed to be superior to execution-based feedback. This creates a circular dependency - if the execution-based labels are noisy (as the paper argues), then the reward model learns to approximate noisy signals. The paper acknowledges this ("some of them might be noisy") but does not rigorously analyze how noise in training labels affects model quality or how the model could possibly exceed the quality of its training signal.

**Questions:**

Please refer to questions raised in weaknesses.

---

> ### Author Response · Authors · 2025-11-23
> **Rebuttal to Reviewer aGbU [1]**
>
> Dear Reviewer aGbU,
>
> Thank you for your insightful comments! And we are grateful that you found our observation is interesting, reported improvements are good and SWE-RM addresses a practical limitation of prior work. We address your comments one by one as follows:
>
> > The paper does not adequately address a critical limitation: verifying program correctness can be as difficult as, or even harder than, executing the program. A reward model must essentially predict execution outcomes across potentially many different code paths and edge cases without actually running the code. This requires the model to simulate program semantics, which is fundamentally challenging and may introduce systematic errors. The paper does not discuss: How the reward model handles corner cases or edge conditions that unit tests would catch. The types of errors the reward model systematically misses (false negatives) or incorrectly flags (false positives).  The claim that execution-free feedback is superior needs much stronger grounding.
>
> We agree that predicting program correctness is fundamentally challenging, and we do not position execution-free verification as a perfect oracle. In our work, we demonstrate that such reward models can outperform noisy unit test verification, where both the reward models and unit test are not perfect. These noisy reward models and unit tests can even be complementary, as shown in our RL experiments with hybrid feedback. While we acknowledge that unit test verification is inherently more reliable and interpretable **when they are well-designed**, we believe that exploring execution-free verification is a worthwhile academic endeavor.
>
> > Whether execution-free verification can ever be truly reliable for complex software engineering tasks. Why we should expect a learned model to generalize to unseen code patterns and bugs. This represents a significant gap in the theoretical justification for the entire approach.
>
> We understand this overall sentiment, but "lacking theoretical justification" or "not truly reliable" **doesn't invalidate the entire approach or direction of reward models for software engineering**. Yes, maybe reward models for SWE tasks are never truly reliable, but LLMs are not truly reliable either for most domains, including chatting, coding, and reasoning -- LLMs keep write bugs, hallucinate, and generate wrong chain-of-thought reasoning, LLMs are never guaranteed to be reliable, while at the same time they are already pretty useful in those areas. For large models pretrained on tons of code, we do believe it can generalize to unseen code patterns and bugs in the future -- Regardless of whether this belief is ultimately correct or not, we think it is valid to pursue such goals as an academic endeavor in this paper.
>
> > While the paper presents extensive ablations, the intuitions behind many design decisions are unclear: Why is a 2:1 positive/negative ratio optimal? Is this simply an artifact of data availability, or is there a principled reason? Why do mixed policies help? What specific distribution shift does this address? The paper states "we believe reward model can still achieve relative good performance by training on a large number of data" (Appendix C.1) when discussing noisy labels, but provides no theoretical or empirical justification for this belief Why are AUC and ECE the right metrics? While intuitively reasonable, there's no formal argument for why these metrics should predict RL performance
>
> Thank you for raising these points. The analysis of how to build a versatile reward model (ratio, mixture etc.) is entirely built upon experimental results, and—as is common in many technical reports—many design choices are guided by empirical ablations rather than strong theories. For the positive/negative ratio and policy mixture, our intention is not to claim theoretical optimality, but to report the configuration that consistently performed best across AUC, ECE, and TTS in controlled comparisons (e.g., Table 1 and Table 3). For example , in Table 1, the 2:1 ratio emerged as the best setting relative to alternatives, rather than being chosen due to principled reason.
>
> Regarding the justification for why AUC and ECE are the right metrics, we first provide an intuitive illustration of why top-1 ranking(TTS) alone is insufficient, and why discrimination and calibration matter in Section 3. In response to your theoretical concern, we have additionally included a more formal analysis in Appendix C of the updated PDF, clarifying how these metrics relate to reward-shaping stability and policy-gradient noise. We would appreciate it if you could check Appendix C and we hope this resolves your concern of why these metrics should be related to RL performance.

---

> ### Author Response · Authors · 2025-11-23
> **Rebuttal to Reviewer aGbU [2]**
>
> > The core contribution appears to be training a larger reward model on more diverse data and identifying AUC/calibration as important metrics. The approach of training a classifier on successful/unsuccessful trajectories and using it for ranking is standard. The paper does not introduce novel algorithms, architectures (beyond using MoE), or training objectives. The identification of additional evaluation metrics, while useful, is incremental rather than constituting a significant methodological advance.
>
> We respectfully clarify that the core contribution of our work is not any major machine learning methodological advance, but **novel insights and discussion on how to define and what metrics to monitor to obtain a good SWE reward models for both TTS and RL.** These insights should be practically useful for developers who train reward models.
>
> > The reward model is trained on trajectories labeled by execution-based feedback, then claimed to be superior to execution-based feedback. This creates a circular dependency - if the execution-based labels are noisy (as the paper argues), then the reward model learns to approximate noisy signals. The paper acknowledges this ("some of them might be noisy") but does not rigorously analyze how noise in training labels affects model quality or how the model could possibly exceed the quality of its training signal.
>
> This is a good point. Our assumption is that a sufficiently large and diverse dataset enables the RM to learn a denoised and generalized correctness signal despite noisy supervision, which is also acknowledged  by Reviewer YTBh:
> > The paper "believe reward model can still achieve relative good performance by training on a large number of data", which is a reasonable assumption (i.e., the model learns a denoised, generalized signal)
>
> Besides this assumption, we also applied data cleaning such as filtering out instances without any successful trajectory  (typically cases affected by over-strict/unfair unit tests or under-specified descriptions) to maintain the highest possible label quality, as described in Appendix D.1 Data Collection. Together with the "denoising" effect, empirically, SWE-RM is able to **reduce errors on the human-curated SWE-Bench Verified benchmark (which serves as a noise-free oracle) to 8% false positives and 19.4% false negatives**, substantially lower than the noise levels of the original SWE-Bench  execution labels of 11% false positives and 28.3% false negatives[1,2]. We think this empirically demonstrates that our RM is able to decrease the errors in execution-based labels even though the RM itself is trained on flawed signals.
> In addition, we would like to note that execution-free feedback offers another critical advantage: unlike execution-based signals that are sparse 0/1, execution-free scores are dense and continuous, allowing the RM to provide finer-grained discrimination among partially correct, near-miss, or suboptimal trajectories. This not only makes feedback more informative but also more reliable for shaping policy updates. Finally, in RL, hybrid feedback leverages complementary failure modes: execution-free and execution-based signals tend to err differently, reducing the probability that both provide the same flawed feedback and thus improving the robustness and stability of the overall reward signal (as reflected in Figure 7). Together, these factors explain why, even when trained on noisy labels, the RM can denoise effectively and enable more stable and reliable TTS improvements and RL training.
>
> [1] Introducing SWE-bench Verified, OpenAI, Aug 13, 2024, https://openai.com/index/introducing-swe-bench-verified/
>
> [2] Are “Solved Issues” in SWE-bench Really Solved Correctly? An Empirical Study, You W. et,al. 2025

---

### Official Review · Reviewer_dWfQ · 2025-11-01

**Soundness:** 3
**Presentation:** 2
**Contribution:** 3
**Rating:** 6
**Confidence:** 3

**Summary:**

This paper proposes SWE-RM, a large-scale, execution-free reward model for SWE agents, and shows it is effective both for test-time scaling (TTS) and as a reward signal in RL. Execution-based verifiers rely on incomplete or noisy unit tests, producing sparse feedback, while execution-free models provide denser, more informative signals. The authors argue that TTS performance alone is insufficient for evaluating verifier quality and introduce AUC (discriminative ability) and ECE (calibration error) as complementary metrics. Extensive ablations explore the effects of dataset size, composition, positive/negative ratios, and policy mixtures. SWE-RM achieves strong TTS gains on SWE-Bench Verified and modest RL improvements when combined with execution-based rewards.

**Strengths:**

- Showing that identical TTS performance can yield divergent RL outcomes is an important empirical finding with implications for verifier evaluation and selection.
- The ablations on data scale, source composition, and context length provide concrete, actionable insights for building robust SWE reward models.
- The model supports very long (256k) contexts so the verifier can score large numbers of trajectories.
- A single verifier that supports both evaluation-time reranking and RL training is practically useful.

**Weaknesses:**

- The narrative sometimes shifts between TTS (used for evaluation/selection) and the execution-free RM (the actual trained model), which can make the pipeline slightly harder to follow. Being explicit about where TTS stops and supervised RM training begins would help.
- Because the paper starts from TTS limitations, it would help to spell out the actual supervised reward-model objective earlier so readers don’t assume TTS is the training signal.
- Evaluation is limited to SWE-Bench Verified, leaving generalization to other SWE tasks or domains untested.
- The model's scale and context requirements may constrain practical deployment.

**Questions:**

- The authors show cases where similar TTS but better AUC/ECE give better RL. Is there evidence that improving calibration directly improves RL, or guidance on thresholds?

---

> ### Author Response · Authors · 2025-11-23
> **Rebuttal to Reviewer dWfQ [1]**
>
> Dear Reviewer dWfQ,
>
> Thank you for your insightful comments! And we are grateful that you found our work has an important empirical finding and provides actionable insights. We address your comments one by one as follows:
>
> > The narrative sometimes shifts between TTS (used for evaluation/selection) and the execution-free RM (the actual trained model), which can make the pipeline slightly harder to follow. Being explicit about where TTS stops and supervised RM training begins would help.
>
> We appreciate the reviewer’s suggestions. We clarify that TTS and the execution-free RM correspond to two distinct but connected stages in our paper: (1) when we discuss the limitation of TTS metric, we provide with some evidence from early trained RM as an example. (2) After showing the arguments of bringing additional metrics of AUC and calibration, we can better assess trained RM's quality. We then begin large scale actual training/ablations.
>
> To make this clearer, we have revised the PDF in §3.2 to explicitly indicate the boundary as highlighted in revised pdf (Line 239 - Line 242)
>
> > Because the paper starts from TTS limitations, it would help to spell out the actual supervised reward-model objective earlier so readers don’t assume TTS is the training signal.
>
> Thank you for the helpful suggestion. To avoid the misconception that TTS might be used as a training signal, we have added an explicit explanation of the supervised reward-model objective in §3.1 (Line 157 - 160). We clarify that the RM is trained solely via standard next-token prediction loss on labeled trajectories, and that TTS is never optimized against, but is used only as an external diagnostic metric for assessing verifier quality. We believe this addition removes ambiguity and improves the overall readability of paper.
>
> > Evaluation is limited to SWE-Bench Verified, leaving generalization to other SWE tasks or domains untested.
>
> | Method                | SWE-bench Live (Lite) | SWE-bench multilingual | Multi-SWE-bench mini | Terminal Bench |
> |-----------------------|------------------------|-------------------------|-----------------------|----------------|
> | Hybrid            | **22.4**               | **35.7**                | **20.0**              | **32.5**       |
> | Execution-free only   | 20.4                   | 33.0                    | 18.8                  | 31.3           |
> | Execution-based only  | 20.0                   | 33.3                    | 18.5                  | 30.0           |
> | Poor Calibrated RM    | 12.0                   | 21.0                    | 10.0                  | 15.0           |
>
> Thank you for raising this point. While our main evaluation focuses on SWE-Bench Verified, we additionally conducted experiments on a broader suite of SWE tasks—including SWE-Bench Live (Lite), SWE-Bench Multilingual, Multi-SWE-Bench Mini, and Terminal Bench—to assess generalization beyond the original domain. As shown in table above, hybrid feedback consistently achieves better RL performance, while execution-free only feedback shows comparable results to execution-based only feedback. And if using feedback from a poorly calibrated RM, the model will also show a significant decrease in other tasks. We have also added this part in the updated PDF §5.2  (Line 496 - 501) and Table 5.
>
> > The model's scale and context requirements may constrain practical deployment.
>
> We appreciate the reviewer’s concern regarding practical usage. While SWE-RM adopts a 30B-parameter Mixture-of-Experts architecture, only 3B parameters are activated per inference, resulting in computational costs comparable to 9B-level models. Regarding context length, the 256k window is not a strict requirement, but rather a capability enabling coverage of long, multi-turn SWE trajectories. In addition, frontier coding-agent policy models such as  Claude (1M context), GPT-5 (200k context), Qwen3‑Coder (256k context) and Kimi‑K2 (256k context) , have pushed to extremely long contexts for precisely the reason of covering very large codebases, traces, and chains of thought. **In our test, a user with 2 A100 GPUs can deploy our 256k model with high efficiency. Therefore, we believe our model is practical for deployment with reasonable hardware**.
>
> > The authors show cases where similar TTS but better AUC/ECE give better RL. Is there evidence that improving calibration directly improves RL, or guidance on thresholds?
>
> Thank you for raising this point. While our empirical results show that reward models with better calibration consistently lead to more stable RL training, establishing a causal, quantitative relationship between calibration levels and RL performance remains an interesting direction for future work. We view deriving explicit calibration–RL thresholds or guarantees as an important direction for future research.

---

### Official Review · Reviewer_YTBh · 2025-11-01

**Soundness:** 3
**Presentation:** 4
**Contribution:** 3
**Rating:** 6
**Confidence:** 3

**Summary:**

This paper proposes to train an execution-free reward model (RM) to address the reliance on the real-world execution feedback for training software engineering (SWE) agents, which is often sparse, unreliable, and hard to scale. The paper's core insight is athat good performance on test-time scaling (TTS), a common metric for verifiers, does not gaurantee good performance when the verifier is used as a reward model for RL. The authors identify 2 crucial properties for a robust RM: discriminative ability (measured by AUC) and confidence reliability (measured by calibration, or ECE). Based on this insight the authors conduct comprehensive experiments to determine how to train a robust RM, by investigating factors like training data sclae, positive-to-negative data ratios, data source etc. The authors train SWE-RM a 30B MoE model, starting with Qwen3-30B-A3B, which achieves solid improvement of pass@1 on SWE-Bench Verified.

**Strengths:**

- The paper's primary strength is its core insight, which is clearly motivated and empirically demonstrated. The finding that test-time scaling (TTS) performance is an insufficient proxy for a reward model's utility in RL  is interesting.
- The experiments are well-executed, a series of ablation is done on data scale, data composition, policy mixture and context length. This provides good guidance and insight for the community to identify the source of gain.
- The writing is exceptionally well-written, logically structured, and easy to follow. The paper also presents strong empirical results when the RM is used as a verifier for TTS and as additional source of signal during RL.

**Weaknesses:**

- Opacity of the "Poor Calibrated RM" (Verifier B): The entire paper's motivation hinges on the comparison between "Verifier A (Good AUC & Cali.)" and "Verifier B (Bad AUC & Cali.)". We are shown they have similar TTS but different RL outcomes. However, the paper never explains how Verifier B was trained or why it has bad calibration. Is it off-the-shelf (together with verifier A) or trained by the authors (if so could authors explain how to train it)? Is it the same as "poorly calibrated RM" in Figure 7?
- There's a slight tension in the paper's premise. The RM is trained on positive/negative labels generated by "execution results with the provided fail2pass test". However, the paper's introduction effectively argues that these very tests are unreliable, sparse, and "in some cases, entirely unrelated to the target issue" . The paper "believe reward model can still achieve relative good performance by training on a large number of data", which is a reasonable assumption (i.e., the model learns a denoised, generalized signal). Still, this is a key point: the RM is trained on the same (flawed) signal it is meant to improve upon. A deeper discussion of this "denoising" effect would be welcome. And the issue of noise is also coupled with the Figure 7 result of execution-only run, where the authors attributes its performance drops to "issues with test noise and coverage".

**Questions:**

- What dataset Figure 4 reports on? I would love to know more details about how the authors compute the ECE, e.g. on which dataset, how many rollouts, sampling config, etc.
- How were the base reward constants (+1.0 for resolve, -0.5 for unfinished, 0.0 for otherwise) chosen? Did you experiment with other values, and how sensitive is the final RL performance (Figure 7) to these specific hyperparameters?
- The data scaling experiment (Figure 6) and the data composition experiments (Tables 1 & 3)  are presented separately. For clarity, was the 100k data point in the scaling experiment trained using the optimal 2:1 ratio and policy/source mix found later? And conversely, were the composition experiments in Tables 1 & 3 conducted at the full 100k data scale, or a smaller one? I'm trying to understand the interplay between data quantity and quality/composition.

For the rest, please see above for the weakness part.

---

> ### Author Response · Authors · 2025-11-23
> **Rebuttal to Reviewer YTBh [1]**
>
> Dear Reviewer YTBh,
>
> Thank you for your insightful comments! And we are grateful that you found our work clearly motivated and empirically demonstrated. We address your comments one by one as follows:
>
> > Opacity of the "Poor Calibrated RM" (Verifier B): The entire paper's motivation hinges on the comparison between "Verifier A (Good AUC & Cali.)" and "Verifier B (Bad AUC & Cali.)". We are shown they have similar TTS but different RL outcomes. However, the paper never explains how Verifier B was trained or why it has bad calibration. Is it off-the-shelf (together with verifier A) or trained by the authors (if so could authors explain how to train it)? Is it the same as "poorly calibrated RM" in Figure 7?
>
> Verifier A and Verifier B were both trained by us during early exploration, not taken off-the-shelf. We take them as two representatives because they have similar TTS (which we originally set as model selection criteria) but different AUC and Calibration. They are trained using the same approach described in §4.1. They differ in training settings such as data sources, policy mixtures, and positive/negative ratios. We have added Table 8 in Appendix D.3 of the updated PDF to summarize these settings along with the "poorly calibrated RM" in Figure 7.
>
> Verifier B’s poor AUC/Calibration mainly arises from its less diverse data composition, policy and imbalanced data ratio to negative data, which our ablations (§4) show to consistently harm AUC and ECE. Our goal in presenting A vs. B is to illustrate that similar TTS does not guarantee similar RL behavior, motivating the need to evaluate verifiers along with discrimination and calibration, not TTS alone.
>
> > There's a slight tension in the paper's premise. The RM is trained on positive/negative labels generated by "execution results with the provided fail2pass test". However, the paper's introduction effectively argues that these very tests are unreliable, sparse, and "in some cases, entirely unrelated to the target issue" . The paper "believe reward model can still achieve relative good performance by training on a large number of data", which is a reasonable assumption (i.e., the model learns a denoised, generalized signal). Still, this is a key point: the RM is trained on the same (flawed) signal it is meant to improve upon. A deeper discussion of this "denoising" effect would be welcome. And the issue of noise is also coupled with the Figure 7 result of execution-only run, where the authors attributes its performance drops to "issues with test noise and coverage".
>
> This is a good point. Besides our assumption that a sufficiently large and diverse dataset enables the RM to learn a denoised and generalized correctness signal despite noisy supervision, we also applied data cleaning such as filtering out instances without any successful trajectory  (typically cases affected by over-strict/unfair unit tests or under-specified descriptions) to maintain the highest possible label quality, as described in Appendix D.1 Data Collection. Together with the "denoising" effect, empirically, SWE-RM is able to reduce errors on the human-curated SWE-Bench Verified benchmark (which serves as a noise-free oracle) to 8% false positives and 19.4% false negatives, substantially lower than the noise levels of the original SWE-Bench execution labels of 11% false positives and 28.3% false negatives[1,2]. We think this empirically demonstrates that our RM is able to decrease the errors in execution-based labels even though the RM itself is trained on flawed signals.
>
> In addition, we would like to note that execution-free feedback offers another critical advantage: unlike execution-based signals that are sparse 0/1, execution-free scores are dense and continuous, allowing the RM to provide finer-grained discrimination among partially correct, near-miss, or suboptimal trajectories. This not only makes feedback more informative but also more reliable for shaping policy updates. Finally, in RL, hybrid feedback leverages complementary failure modes: execution-free and execution-based signals tend to err differently, reducing the probability that both provide the same flawed feedback and thus improving the robustness and stability of the overall reward signal (as reflected in Figure 7). Together, these factors explain why, even when trained on noisy labels, the RM can denoise effectively and enable more stable and reliable TTS improvements and RL training.
>
> [1] Introducing SWE-bench Verified, OpenAI, Aug 13, 2024, https://openai.com/index/introducing-swe-bench-verified/
>
> [2] Are “Solved Issues” in SWE-bench Really Solved Correctly? An Empirical Study, You W. et,al. 2025

---

> ### Author Response · Authors · 2025-11-23
> **Rebuttal to Reviewer YTBh [2]**
>
> > What dataset Figure 4 reports on? I would love to know more details about how the authors compute the ECE, e.g. on which dataset, how many rollouts, sampling config, etc.
>
> For Figure 4 and 5, the evaluation is the same as the setting stated in Appendix C.1 (original pdf). The calibration is reported on the SWE-Bench Verified dataset, where ECE is computed by
>
> $$\mathrm{ECE} = \sum_{m=1}^{M} \frac{|B_m|}{n}\Bigl|\mathrm{acc}(B_m) - \mathrm{conf}(B_m)\Bigr|,
> \mathrm{conf}(B_m) = \frac{1}{|B_m|}\sum_{i \in B_m} r_i,
> \mathrm{acc}(B_m) = \frac{1}{|B_m|}\sum_{i \in B_m} c_i$$
> In Figure 4, we follow common practice to divide confidence into 10 bins (M = 10) and calculate accuracy and confidence within each bin ($B_m$). We sample 32 rollouts of Qwen-Coder-Max on Swe-Bench Verified, resulting  in 32 * 500 = 16000 total trajectories using Openhands with 100 max iterations and temperature = 1.0. We have incorporated these details into the revised pdf §3.2 (Line 232-234) and Appendix D.1 (Line 933-938).
>
> > How were the base reward constants (+1.0 for resolve, -0.5 for unfinished, 0.0 for otherwise) chosen? Did you experiment with other values, and how sensitive is the final RL performance (Figure 7) to these specific hyperparameters?
>
> Our choice of the base reward constants follows common practice in rule-based RL such as DeepSeek-R1 and Qwen3-Coder, where correct/incorrect (execution) outcomes are typically mapped to a 0/1 score. This also matches the reward model’s output scale (a continuous score between 0 and 1), enabling the two sources of feedback to be combined seamlessly.
>
> For the penalty on unfinished trajectories, we experimented with several alternatives. A more aggressive penalty (-1.0) increases variance and destabilizes learning, as the policy struggles to distinguish “unfinished” trajectories from genuinely incorrect ones. Conversely, a small penalty (-0.25), we observed persistent over-long unfinished trajectories during RL. Based on this trade-off, we selected (-0.5) as a balanced penalty to maintain stable training dynamics. We then focus on analyzing how the reward model’s continuous scores influence RL performance, since this component is the primary factor driving the differences studied in the paper.
>
> > The data scaling experiment (Figure 6) and the data composition experiments (Tables 1 & 3) are presented separately. For clarity, was the 100k data point in the scaling experiment trained using the optimal 2:1 ratio and policy/source mix found later? And conversely, were the composition experiments in Tables 1 & 3 conducted at the full 100k data scale, or a smaller one? I'm trying to understand the interplay between data quantity and quality/composition.
>
> Thank you for pointing out the need to clarify the ablation setting details, which can greatly help understanding the reward model training dynamics. For data scaling experiments, we all use the optimal 2:1 ratio. We also adapt the 2:1 ratio for the later policy/source mixture. However, for composition experiments, the data scale is smaller, as we don't have 100k data for each setting. For table 1, we use 25k as the total data size, while for table 3, we use 20k as the total data size.

---

### Public Comment · ~Wasi_Uddin_Ahmad1 · 2025-11-25
**Questions regarding finetuning setup**

This is a pretty interesting work. For reproducibility, I have a few questions.

1. In page 18, Table 7 says the work used Qwen3 30BA3 model. Is it this one - https://huggingface.co/Qwen/Qwen3-30B-A3B? Or, some coder/instruct only version has been finetuned?
2. How the authors found out the learning rate (7e-6 -> 7e-7)?
3. Did the authors use weight decay?

Another question: for evaluation, which LLM agent is used to get the "K" trajectories? For training data, the paper says, it used Qwen3-Coder and Claude-4. So, curious, which LLM is used for evaluation.

---

> ### Author Response · Authors · 2025-11-28
> **Response**
>
> Thank you for your questions.
>
> 1. Our backbone is a model with Qwen3-30B-A3B architecture and further pre-trained on coding domain for better coding ability. To maintain double-blindness, we describe it as an anonymized internal variant and we will make it more clear in the final version. We hope this could resolve the concern while preserving the anonymity required for the review process.
>
> 2. This learning rate, along with decay, is iterated in our fine-tuning steps, where a smaller learning rate will require more epoches to train. And we choose the smallest learning rate that can achieve the best result at one epoch. We also have a weight decay of 0.1.
>
> 3. For the test-time scaling experiments, the "k" trajectories are sampled using Qwen3-Coder-Max, Qwen3-Coder-Flash, OpenHands-LM-32B under three distinct settings. Under each setting, we sampled 32 trajectories for 500 instances in SWE-Bench Verified. And in each table, we also have a model description that indicates which model is used to sample the trajectories (evaluation).

---

### Author Response · Authors · 2025-11-27
**Response to Reviewers**

Dear Reviewers,

I hope this message finds you well. As it has been several days since we posted the responses, we want to ensure we have addressed all your concerns satisfactorily. If there are any additional points or feedback you would like us to consider, please let us know. Your insights are invaluable to us, and we are eager to address any remaining concerns to improve our work.

Thank you for your time and effort in reviewing our paper.

Best Regards,

Authors

---

### Meta-Review · Area_Chair_nRRc · 2026-01-07

**Summary:**

Given the details below that paper would still be borderline but more on the accept side as key issues raised in the reviews were largely addressed with added theory, more training details, broader evaluations, and noise/cleaning evidence. Remaining weaknesses are mainly novelty, architectural ablations, and lack of rigorous guarantees. While one reviewer also has reservations regarding the conceptual basis of the work, I think the direction the work takes might spin off interesting works that might combine the different approaches.

**Reviewer Concerns:**

Reviewer YTBh
- Clarity about "Verifier B" (training, why poorly calibrated, relation to Figure 7)
  - Mainly addressed by clarifying the training setting (Table 8 added); whether other verifiers might have different properties/effects is unclear
- Tension in the paper's premise: the reward model (RM) is trained on noisy execution labels it aims to improve over
  - Addressed but also highlights and underlying assumption which would have benefitted from a more extensive discussion
- ECE computation details (dataset, rollouts, bins, configs)
  - Addressed/clarified
- Reward constants choice and sensitivity
  - Addressed by clarifying the tested parameter ranges
- Interplay between data scale and composition (ratios, mixes, scales used)
  - Addressed (2:1 ratio used; composition experiments at 25k/20k); where the ratio comes from is clarified in a response to a different reviewer

Reviewer dWfQ
- Pipeline clarity (boundary between TTS and supervised RM training)
  - Addressed by clarification and explicit revisions in Section 3.2.
- Limited evaluation beyond SWE-Bench Verified
  - Addressed by inluding additional experiments on SWE-Bench Live (Lite)
- Practicality concerns (model scale/context)
  - Partially addressed but details regarding the final claims are missing
- Evidence that calibration directly improves RL
  - Partially addressed by noting empirical stability, but no causal relationships are identified (considered as future work).

Reviewer aGbU
- Fundamental reliability of execution-free verification (corner cases, systematic errors, generalization)
  - Partially addressed (authors argue regarding complementarity and utility but no detailed error taxonomy or guarantees are provided)
- Lack of theoretical justification for generalization
  - Not fully addressed (framed as an academic direction; belief-based); also argues that certain issues from LLMs are simply inherited
- Design intuitions (2:1 ratio optimality, mixed policy rationale, noisy-label robustness)
  - Partially addressed (empirical insights with limited principled justification)
- Why AUC/ECE are the right metrics
  - Addressed by adding a formal analysis in Appendix C linking to RL dynamics but would require a careful review
- Contribution novelty (incremental vs. methodological advance)
  - Authors position contribution as practical insights; how to judge this is a matter of taste
-  Circular dependency on noisy execution labels
  - Partially addressed (data cleaning; empirical error-rate reductions; hybrid signal rationale; no rigorous noise-model analysis)

Reviewer nzG1
- Missing theory linking TTS/AUC/ECE to RL stability
  -   Addressed by adding a formal analysis but would require a careful review
- MoE choice and lack of architectural ablations
  - Partially addressed (pragmatic choice aligned with SOTA) so concerns likely persist
- "Execution-free" despite execution-derived labels
  - Addressed by clarifying the standard paradigm of being execution-free at inference while training uses execution-derived labels
- Context-length gains and trade-offs
  - Addressed

**Reviewer Scores:**

* Reviewer YTBh might have incerased their score as their concerns were well addressed
* Reviewer dWfQ would likely keep their score since concerns regarding the relation of calibration and improvements in RL remain although the other concerns were well addressed
* Reviewer aGbU might have increased their score but would still remain critical about the paper since concerns regarding principled justification and novelty were not fully addressed
* Reviewer nzG1might have increased their score since most concerns were well addressed

---

### Decision · Program_Chairs · 2026-01-26

Accept (Poster)